# New approach to improve power consumption associated with blockchain in WSNs

**Maytham S. Jabor** \*, **Aqeel Salman Azez, José Carlos Campelo, Alberto Bonastre Pina**

Instituto ITACA, Universitat Politècnica de València, Valencia, España

\* mayaz@doctor.upv.es

## Abstract

Nowadays, Wireless Sensor Networks (WSNs) are widely used for collecting, communicating, and sharing information in various applications. Due to its limited resources in terms of computation, power, battery lifetime, and memory storage for sensor nodes, it is difficult to add confidentiality and integrity security features. It is worth noting that blockchain (BC) technology is one of the most promising technologies, because it provides security, avoids centralization, and a trusted third party. However, to apply BCs in WSNs is not an easy task because BC is typically resource-hungry for energy, computation, and memory. In this paper, the additional complication of adding BC in WSNs is compensated by an energy minimization strategy, which basically depends on minimizing the processing load of generating the blockchain hash value, and encrypting and compressing the data that travel from the cluster-heads to the base station to reduce the overall traffic, leading to reduced energy per node. A specific (dedicated) circuit is designed to implement the compression technique, generate the blockchain hash values and data encryption. The compression algorithm is based on chaotic theory. A comparison of the power consumed by a WSN using a blockchain implementation with and without the dedicated circuit, illustrates that the hardware design contributes considerably to reduce the consumption of power. When simulating both approaches, the energy consumed when replacing functions by hardware decreases up to 63%.

**Data Availability Statement:** All relevant data are within the paper and its Supporting Information files.

**Funding:** The author(s) received no specific funding for this work.

## Introduction

Wireless sensor networks (WSNs) are multi-hop self-organizing network systems that consists of many low-cost wireless sensor nodes (SNs) connected through wireless communications. The purpose lies in supportively perceiving, collecting, and processing the information of pre-defined variables in a monitoring area and transmitting it to a base station (BS). As a significant part of the Internet of Things (IoT), WSNs play an important role in many applications, such as medical and health related-treatment, national defense, environmental monitoring, and smart homes. In most cases, the SNs are operated by a battery and cannot be recharged. Moreover, the SNs may be positioned in hard-to-reach or inaccessible environments and are

**Competing interests:** The authors have declared that no competing interests exist.

anticipated to stay operating for several months or years. Therefore, mechanisms to reduce the energy consumption of these nodes and maximize the network life cycle have lately attracted the attention of many researchers [1].

In addition, most of WSNs are defenseless against security threats and pose several security challenges [2]. Therefore, security in WSNs and IoT is a crucial and not easy problem [3]. All security mechanisms require certain resources for their implementation, such as code space, data memory, and energy to provide power to sensor, but these resources are very limited in small wireless sensors [4].

One type of security technology is a blockchain (BC). The security of blockchain systems is vital for potential users [5]. BCs have attracted researchers' attention, as they believe that this technology will bring about extraordinary changes and opportunities in the world of industries. BCs are seen as a very robust technology for resolving trusted communications in a decentralized manner. This technology was introduced in 2008 by Nakamoto [6], and has been circulated by the encryption postal group. The basic power of BC is decentralized, allowing direct transactions point-to-point. It is possible to use this approach in distributed systems, since trust is not required for nodes to perform transactions, because a third party is no longer needed in a blockchain [7]. The definition of blockchain is a chain of valid transaction blocks, wherein each one contains the hash of a previous block in blockchain. After verifying the transaction, it is broadcasted to network and added to each blockchain copy. A blockchain is fundamentally a decentralized, distributed, shared, and immutable database ledger that stores registry of assets and transactions across a peer-to-peer (P2P) network [8]. Clearly, the advantages of BC technology can be listed as follows [9]:

- **Security**: Neither node nor anyone else, except transmitter and receiver can have access to data transmitted via BC.

- **Removal** of intermediaries: The peer-to-peer nature of BC requires no intermediaries.

- **Immutability**: Nothing on the BC can change, and any confirmed transaction cannot be altered.

- **Permanence**: A public BC acts as a public ledger. If the BC remains active, data will be accessible.

- **Speed**: According to [10], transactions are quicker than centrally controlled ledger. because Blockchain removes any third-party intervention between transactions and removes mistake and make system efficient and faster.

Generally, a block contains a timestamp, key data, previous and current block hash, and a set of transactions, which is a piece of data representing an operation that some users want to carry out. Fig 1 illustrates the structure of block. When executing the transaction, it is hashed into a code and then broadcasted to each node. Since thousands of transaction records can be contained in block of each node, the BC uses the Merkle tree function to generate final hash value, which is also the root of the Merkle tree. This final hash value will be recorded in the block head the hash of the current block, with the Merkle tree function, computing resources, and thus the data transmission can be greatly minimized [11]. Blockchain provides authentication, confidentiality, and integrity through its distributed ledger technology. The distributed ledger is a shared database that is maintained by a network of computers. Each computer in the network stores a copy of the ledger, which is updated whenever a transaction occurs. This ensures that all copies of the ledger are identical and up to date. The authentication process in blockchain is based on digital signatures and cryptographic hashes. Digital signatures are used to verify the identity of the sender and receiver of a transaction, while cryptographic hashes

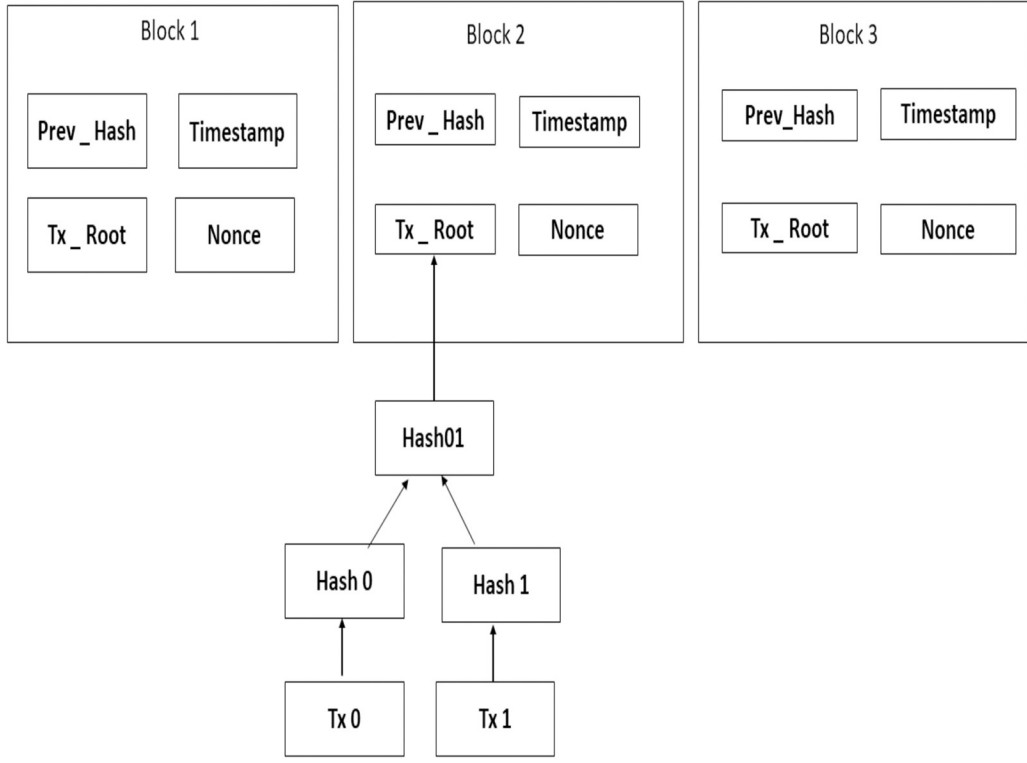

**Fig 1. Blockchain structure.**

are used to ensure that the data stored in the blockchain has not been tampered with or altered in any way. The confidentiality of data stored on the blockchain is ensured through encryption. All data stored on the blockchain is encrypted using public-key cryptography, which makes it impossible for anyone other than those with access to the private key to view or modify it. Finally, blockchain provides integrity by ensuring that all transactions are immutable and cannot be reversed or modified once they have been added to the chain. This ensures that all transactions are valid and trustworthy, as any attempt to alter them would be immediately detected by other nodes in the network [12].

Adopting a energy-intensive and computationally intensive Blockchain technology across all participating nodes of the wireless sensor network to maintain data integrity and privacy will not be easy, as sensor nodes have critical constraint that must be addressed. Most researchers have agreed that WSN nodes/IoT nodes have 3 major limitations, as follows: battery capacity, computing hardware [13,14], and memory. This will create complexity in achieving security features in such systems. Hence, they would not be fast enough to carry data securely in real-time [15,16]. The above-mentioned points represent the challenges faced by WSNs in general, and during the BC technology implementation process in particular.

From an initial point of view, the main cause of power consumption is the process of transmitting and receiving data; hence, the greater the volume of data, the higher the power consumption, because the transmission and reception time will be longer. In addition, the longer the distance between transmitter and receiver, the higher consumption. Moreover, data processing is also an important factor in energy consumption. Data processing includes the following:

- Creation block

- Mining (obtaining a consensus)

- Hash generation

- Creation of encryption keys

- Encryption process and using complicated algorithms

Therefore, the first solution to save energy is to minimize the size of transmitted data by compressing the data. Developments in data compression algorithms have become indispensable in multimedia and communication applications. As is known, there are two main types of compression algorithms: lossless compression and lossy compression. Lossless compression is used for applications that require an exact reconstruction of original data, while lossy compression is used when user can tolerate some differences between the original and reconstructed representations of data.

Due to the limitation of memory and processor speed, many compression algorithms are unsuitable for embedded real-time environments, mainly on the SN [17,18] due to the size of the algorithms; for example, the size of bzip2 is 219 KB and the size of LZO is 220 KB [19]. Therefore, to overcome these problems, energy-saving compression technologies have become a requirement [20,21] being imperative to design an algorithm capable of low-complexity and small-scale data compression for sensor networks [19].

Compressive sensing (CS) is a recently introduced simultaneous data sensing and compression mechanism that performs by sampling randomly at a sub-Nyquist rate. In this situation, the sample time signals can be totally recovered when they are adequately sparsely dispersed in the frequency domain [22]. In addition, from the information security point of view, compressed sensing has received much attention due to the fact that compressed sensing can be regarded as a cryptosystem to attain simultaneous sampling, compression, and encryption when maintaining the secret measurement matrix [23]. This way, CS can provide secrecy if the sensing matrix is changed for each measurement [24]. Thus, this type of cryptosystem can be compared with a One Time Pad system as the private key to encrypt and then decrypt a message is used only once time. In this work, it is aimed to substitute the sequence of random sampling by a deterministic chaotic sequence, as the latter mimics or approximates randomness in only few steps. This effective approach to semi-random sampling across chaotic sequences first emerged in the study by [25].

Besides that, the mathematical theory of chaos can help us in this work. A chaos system is a nonlinear system governed by deterministic equations that has an unstable structure, so that its output acts randomly in only a few steps under particular initial conditions and control parameters [26]. Due to the many intrinsic properties of chaos, such as sensitivity to initial conditions, broadband and orthogonality, and the fact that chaotic signals can be created by low power, low cost, and small area electronic circuits, the use of chaos in communication and engineering has gained a great deal of attention [27–29]. Furthermore, since chaos is just a deterministic equation [26], it is possible to implement chaos on hardware very cheaply. This is because the integration center is able to generate a deterministically transmitted chaotic sequence when the parameters of chaos are known, whereas with random sampling, unstandardized sampling time instants should be sent to the integration center with the time of arrival data [26].

Chaos-based hardware can be employed to optimize system performance by providing a variety of efficient methods to process data. This is because chaotic systems are able to process data faster than traditional methods, leading to improved performance. In addition, increased security by implementing chaos in hardware can help protect against malicious attacks by making it more difficult for attackers to reverse. This is because chaotic systems are highly unpredictable and difficult to model. Finally, reduced cost implementing the chaos in

hardware can reduce the cost of a system by eliminating the need for expensive components such as processors and memory chips. This is because chaotic systems require fewer components, which can lead to cost savings [30,31]. So, CS by chaos achieves a cryptosystem simultaneous to data compression. The chaotic system parameters are used as secret key in constructing measurement matrix and masking matrix [32,33].

Therefore, this works studies the reduction of energy consumption of WSNs while ensuring security thanks to blockchain technology. This solution is based on a dedicated circuit design that works on the basis of chaos theory's principle of the compressive sensing to generate a hash value, data compression and data encryption. The use of hardware-implemented chaotic operations for hash computation is well suited. It provides the recommended statistical properties of the generated random numbers using deterministic formulas, resulting in reduced hardware complexity while preserving reliability compared to probabilistic methods for hardware random numbers generation. In addition, fixed-point hardware is possible when using deterministic formulas, which means further reduction in hardware complexity [34]. As a result, there is no longer necessary to add an algorithm to encrypt data which consumes a lot of resources in terms of time and energy.

To demonstrate the advantages of our hardware approach, we will measure and compare the power consumption and WSN lifetime with respect a WSN provided with traditional blockchain functionality without our dedicated circuit. Moreover, the results will also be compared to a WSN with blockchain implementation and CS techniques implemented in software.

The rest of the paper is organized as follows. Section 2 contains the state-of-the-art of research focused on solutions for minimizing energy consumption in WSN with blockchain. Section 3 presents the proposed system and implementation of the dedicated circuit using a field-programmable gate array FPGA. Section 4 presents the experiments and results. Finally, Section 5 contains the conclusions and future work.

## State of the art

This section reviews the most relevant research that has provided solutions to minimize energy consumption in BC implementations for WSNs and presents the most important aspects that researchers have achieved in this field.

In [35], a secure framework is presented using a smart contract (Ethereum) to choose a cluster-head (CH), which can be combined with the blockchain model for mapping a cluster head. They obtained low energy consumption results when compared to existing WSN network approaches. In addition, they ensured that the data were securely transferred from the sensor node (SN) to sink by cluster-head and integrity of the decentralized database. This model was divided WSN nodes into clusters, each with a number of nodes, one of them being a master node that receives a large amount of data. The remaining nodes in the network (cluster) perform several tasks at the same time. One of them is to send the same information to the master node and mine it directly on the blockchain. The master node, on the other hand, oversees the authenticity of the obtained data by comparing it to the data stored on the blockchain. These researchers provided a new vision of security by utilizing many of the benefits of the blockchain in order to avoid most of possible security problems. In the encryption process, the researchers used public and the private key. The main drawback of this process is the high energy consumption in encrypting and decrypting. Also, the process of verifying transactions occurs in the master node and other nodes do not interfere. This represents one of the issues faced by wireless networks that would result in traffic congestion.

In [36], the use of Blockchain technology was proposed to defend against malicious attacks in IoT networks to transfer wireless energy using the consensus algorithm. The authors

provided a lightweight consensus algorithm based on delegated proof-of-stake (DPoS), to provide substantial energy loss reduction with minimal operating overhead. To design this system, they proposed a new distributed and secure wireless power transfer architecture that included two planes, including energy plane and blockchain plane. In blockchain plane, the devices static energy transmitters acts as miners and responsible for managing blockchain instead of establishing blockchain in the entire network, so that the energy-constrained smart devices had less overhead. The results obtained from their proposal includes energy loss reduction, due to the choice of high-power transmitters to manage the blockchain, which resulted in the time taken for the specified power transmitters to reach consensus being shorter. Furthermore, the probability of a transaction confirmation failure among the specified power transmitters was less. This confirmed the potential energy security gains in terms of energy loss using the DPoS-based blockchain. The researchers presented a strategy to reduce power loss due to the different traffic flows, but the size of the data still presents a power consumption concern, but the amount of data still presents a power consumption concern. After each round of the consensus process, the reward method is adopted. This can encourage some nodes to hold the bids while other nodes do not.

In [37], the red deer algorithm (RDA)-based clustering technique was presented with blockchain—enabled secured data transmission, which was named RDAC-BC. The RDAC-blockchain technique was first initialized, and then the derived fitness function was used to create clusters. The fitness function in the RDAC-BC model was only based on the capacity, power density, node density, distance to nearby nodes, and distance to the sink. Following the completion of the cluster-head election process, a blockchain -based encrypted communication process was initiated. The experimental validation of the RDAC- blockchain technique was evaluated in a number of ways, and the findings were compared to the established methods. This technique performed better energy efficiency, obtained minimum energy consumption, and exhibited a higher network lifetime when compared to other algorithms, such as the genetic algorithm, and Ant-Lion optimization (ALO) and Grey-Wolf optimization (GWO) algorithms. Energy consumption remained huge due to the large volume of energy consumption in the data transmission and reception processes, as well as in processing. This represented a load on the processor of the node, which in turn also resulted in the nodes consuming more energy.

In [38], a proposal to combine blockchain technology with a multiple-collaborative base stations to increase the energy efficiency of WSNs was presented. This proposal can be summarized as a energy-efficient clustering approach based on blockchain technology for WSN networks. In this model, each base station maintained a blockchain to provide consistency of the stored data and network lucidity. This allowed the user to access the status of the nodes that allow it. User could also verify data reliability when multiple base stations in network shared the same data using blockchain technology. All the messages sent between the nodes, users, and the base stations were considered as transactions. Although this proposal provided improved energy efficiency using the clustering approach, it did not provide a solution to reduce the power consumption caused by the volume of data transmitted. In addition, this approach suggested that a cluster key is generated for each group instead of creating keys for each node of the cluster. The generation of keys and hash by software required high power consumption.

In [39], to minimize power consumption and enhance battery life, a load-balancing multi-hop (LBMH) routing and private communication blockchain platform was developed and tested using a constant bit rate. In this system, it was attempted to design a private blockchain in order to secure data. In this framework, the sink node was responsible for authenticating the participating sensors nodes and it stored the node ID, along with the anonymized

blockchain node IDs. To ease the cluster-heads while retrieving the public keys it also stored all of the public keys of all of the participating SNs. All of the cluster-head updated the IDs and private keys of the blockchain to the sink. During the route discovery from the cluster-head of the source to the sink node, the sink stored the reverse route information to the source. Hence, the sink did not need to re-discover the route to the cluster-head again, as long as the routes remain valid. The researchers used a constant bit rate (CBR) mechanism to reduce power consumption, and the nodes used were divided into two types: cluster heads, and sensor nodes. Cluster heads have more energy than the second type. The researchers used the SHA256 algorithm, which is complex and therefore energy-intensive. This will represent a burden on the network.

In [40], linear network coding (LNC) for WSNs was introduced with blockchain -powered IoT devices to increase the efficiency in terms of the number of live nodes, packet distribution ratio, and optimized residual energy. The architecture of the proposed model included the selection of an appropriate cluster head using K-means, encoding using the LNC, and security through the blockchain network. In this model, the gateway was responsible for data reception and forwarding to the blockchain network. The WSN network and gateway were connected through the base station. The sensor data were inserted into the blockchain network using the consensus mechanism. The application layer retrieved data from the blockchain network and performed the data analytics to initiate the appropriate actions in the applications of the smart cities. Researchers used the LNC encryption algorithm to increase the WSNs efficiency in terms of the live node number. In addition to the issue of data size in its transmission and the associated consumption of energy, the researchers have not addressed complexity challenges in the blockchain.

In [41], a blockchain-based protocol was suggested for WSNs to achieve service immutability, availability, and network transparency by using a cooperative multiple-base–stations network to minimize the likelihood of network failure triggered by any attack on the BS and reduce energy consumption. It is attributed that most of the energy is lost in transmitting data to base station. This research presents solutions to reduce energy consumption in sensor networks with blockchain technology. These solutions consist of proposing protocols to balance energy consumption between network nodes and this research suggested using cluster heads with special specifications to extend the network lifetime. In this proposal, the encryption process is accompanied by high energy consumption and computing load. In addition, there is a high energy consumption due to the exchange of keys in the encryption process and in the software generation of the hash. Finally, data transmission still costs high energy consumption.

Table 1 compares the approaches according to their objective, the mechanism for achieving it and the security methods.

To conclude this section, it is worth noting that none of the previous studies offer improvements in data volume reduction or the blockchain complexity in hash creation, which are one of the most important challenges for energy consumption. None of the studies evaluated the benefits of hardware compression and lightweight encryption improvements. These techniques could bring new opportunities for the application of blockchain in resource-constrained systems.

## Proposed system

In this section, the dedicated circuit to provide a solution when implementing a blockchain in a WSN is described. This circuit will be in responsible for performing data compression and security functions (blockchain related functions) to reduce the load on node processors and

**Table 1. Comparison of the proposed survey models.**

| Reference | Year | Aim of approach | The mechanism | Security |
|---|---|---|---|---|
| [35] | 2020 | Low energy consumption | Use a smart contract (Ethereum) to choose a cluster-head (CH) | Use public and the private key, |
| [36] | 2019 | Energy loss reduction | Use of a lightweight consensus algorithm, (DPoS) | Not studied |
| [37] | 2020 | Energy efficiency | Red deer algorithm (RDA)-based clustering technique RDAC-BC | Not studied |
| [38] | 2020 | Improvements in the energy efficiency and security to the WSN | Clustering approach | Use public and the private key |
| [39] | 2019 | Reduce power consumption | Use a constant bit rate (CBR) mechanism | Public key |
| [40] | 2018 | Efficiency improvements in terms of the number of live nodes | Selection of an appropriate cluster head using K-means | Linear network coding (LNC) |
| [41] | 2021 | Reduce energy consumption | Cooperative multiple-base–station | Use public and the private key |

reduce energy consumption. The compression process uses signal compression technique with chaos to select samples, considering encryption at the same time, while the authentication operations include generating the hash.

## Operation of the Blockchain-WSN with the dedicated circuit

This subsection describes the basic operation of the blockchain-WSN on which we will evaluate the feasibility of our proposal, i.e whether our circuit makes possible to use BC technology in WSN. So, the designed circuit will be included in each of its nodes to obtain the results. Basically, the WSN architecture follows the well-established proposals in this area. This way, it follows a structure based on clusters, with cluster heads and sensor nodes. Several routing protocols can be used [42], especially those that generate less complexity and save energy. In this sense, it is suggested a protocol such as Leach to select cluster heads dynamically [43–45], as shown in Fig 2. This routing protocol was selected for our evaluation.

The blockchain paradigm on the WSN network is proposed as a communication procedure among cluster heads, as well as between cluster heads and the base station in a WSN. Data from sensor nodes is sent encrypted and compressed to the cluster head which will add it to the blockchain.

As explained in [44], when nodes join the network or when changes are proposed to present blockchain frameworks in order to establish the blockchain P2P network, the topology building is connected to peer discovery and neighbor selection. The node randomly selects a number between 0 and 1 during the cluster creation stage. Then it compares this number to the threshold value t(n), and if the number is smaller than t(n), it advances to the cluster head position for this round. Otherwise, it becomes a common node. The cluster-heads CHs create a list of the associated sensor IDs to begin data collection.

The sensor nodes send the data compressed and encrypted to the cluster-head using the compressive sensing. When the cluster-head receives this compressed and encrypted data it waits until it collects all the values from its associated nodes and then builds a block of at most 1000 bits to reduce the processing load in cluster-head and reduce the time to send and receive data. If the number of associated nodes of CH is N, the cluster head waits until it receives the N readings, one from each node in its domain. If the cluster-head receives N readings, it assumes that the data is ready to form a block, as all associated nodes have already sent their data. However, if one or more of nodes fails to send its reading for any reason, the cluster head will form a block after a specific waiting time. If the data (block) are sent by another CH, it is

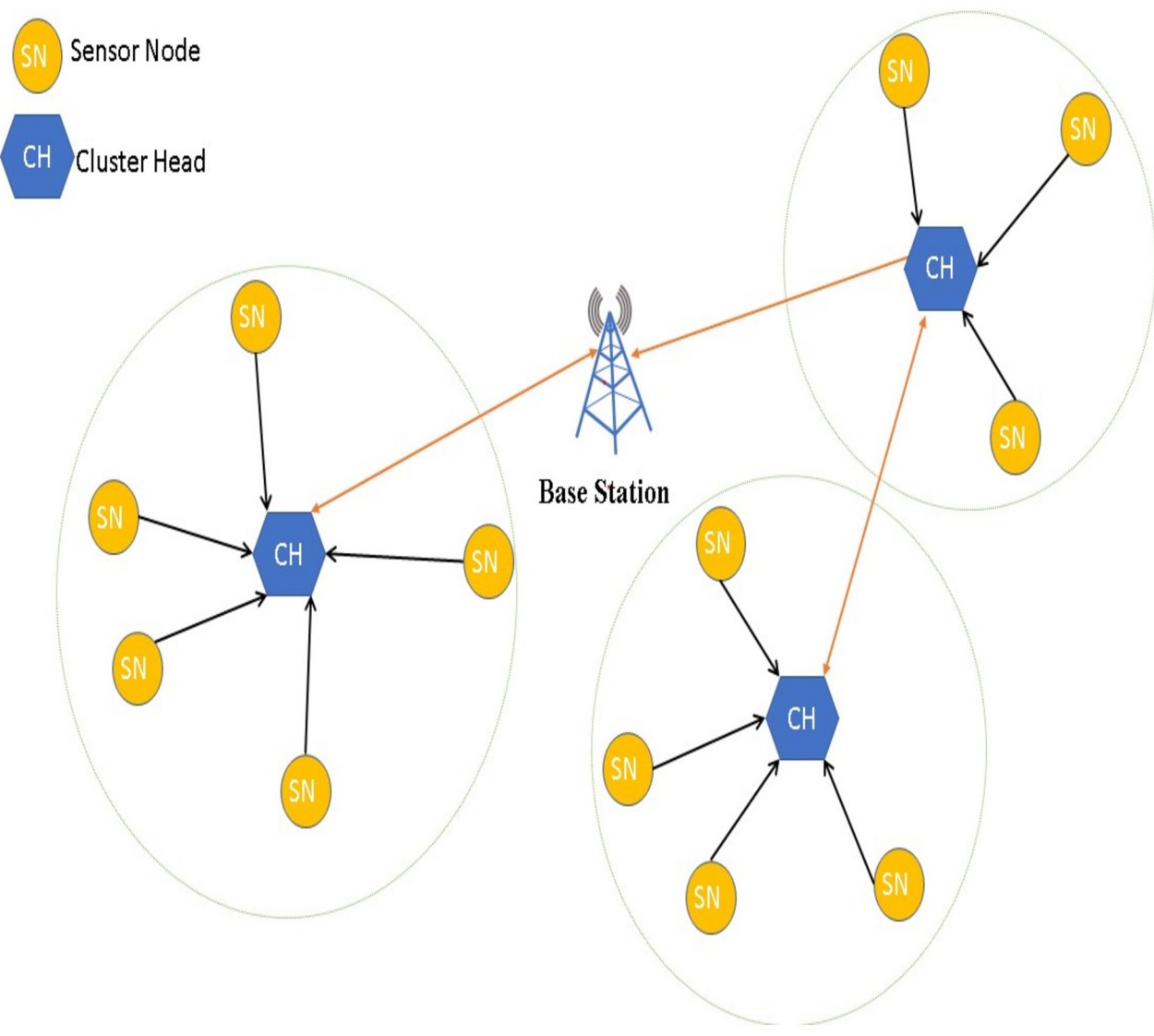

**Fig 2. Network topology [44].**

transmitted without any other operation because this block has already been generated previously, as shown in Fig 3.

The block information will not be considered as reliable unless the majority of the network partners decide its authenticity. In the authentication process, a considerable group of the network members have to recognize the current hash of the block and the previous hash keys, which were included in the block. Each block contains data and initial parameters used in the chaos equation to generate the hash. If the network nodes agree by majority (50%+1) it means that the block is authentic and will be added to chain. That happens when cluster-heads exchange acceptance/rejection messages among each other. Each node in the network has its own ledger to store the authentic blocks. Hence, each node is able to check if the current block has the necessary information about the previous circulating blocks in the network. Inserting a block in the network is like a bid that the network member has to make, as this bid has to wait for the decision of other members to accept or reject it.

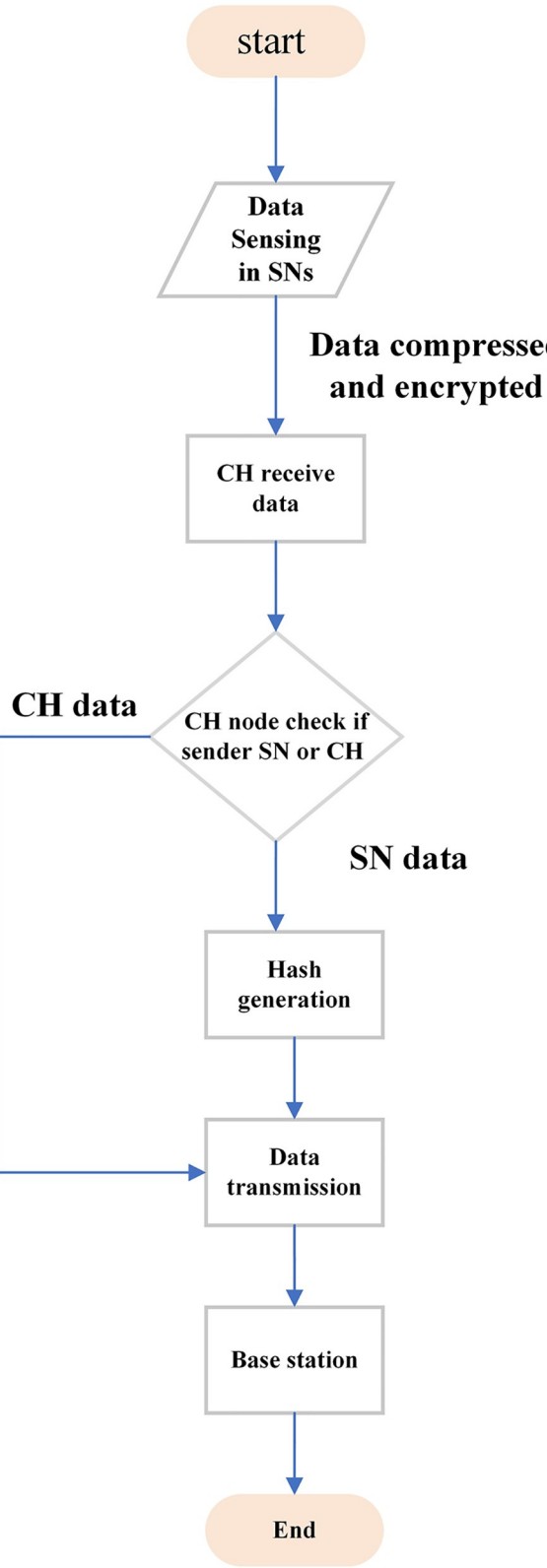

**Fig 3. Flowchart of the data flowing into the network.**

The cluster head bids a new block of information, and the well-known consensus algorithm Proof Of Stake (POS) is used for authenticity [46]. To prevent bids from overlapping, the base station creates a schedule of bids sent by the cluster-heads and works according to the principle of First Come First Serve (FCFS) instead of the mechanism of traditional POS. The consensus algorithm is applied to the consensus depending upon the proposed authentication criteria. To select the CH that creates a new block, CHs send offers to BS. Here, BS creates a schedule of bids sent by CHs and works according to the principle of FCFS. This is to ensure equality among cluster headers for creating a new block and to avoid the control imposed by some nodes in creating blocks. The consensus algorithm is applied on a consensus basis according to the proposed authentication criteria. After creating a bids table in Bs, the Bs send a request for all CHs to listen to the CH that wins a bid, which sends its block for other CHs to validate the block.

The cluster head, which creates a new block, will send it to all cluster-head node neighbors. Each cluster head node transmits the new block to all of the Sensor nodes in its cluster. As a result of the second stage, the current version of the blockchain will be stored on all of the nodes. This is because each node is allowed to be a cluster-head in other rounds.

Blockchain security is based on two phases. The first comprises encryption phase, based on compressed sensor. The second comprises transaction authenticity. In the authentication phase, as shown in Figs 4 and 5, the dedicated circuit generates a hash value, and it is also responsible for authenticating the new block. Fig 4 shows that when Sensor node senses the data, the circuit creates a chaotic sequence, depending on the parameters, to start the sampling process to simultaneously compress and encrypt data. Upon receiving the compressed and encrypted data, the Tx block is generated by cluster-head, produces the chaotic hash code from the dedicated circuit, and then transmits the control parameters within the new block, as shown in Fig 4.

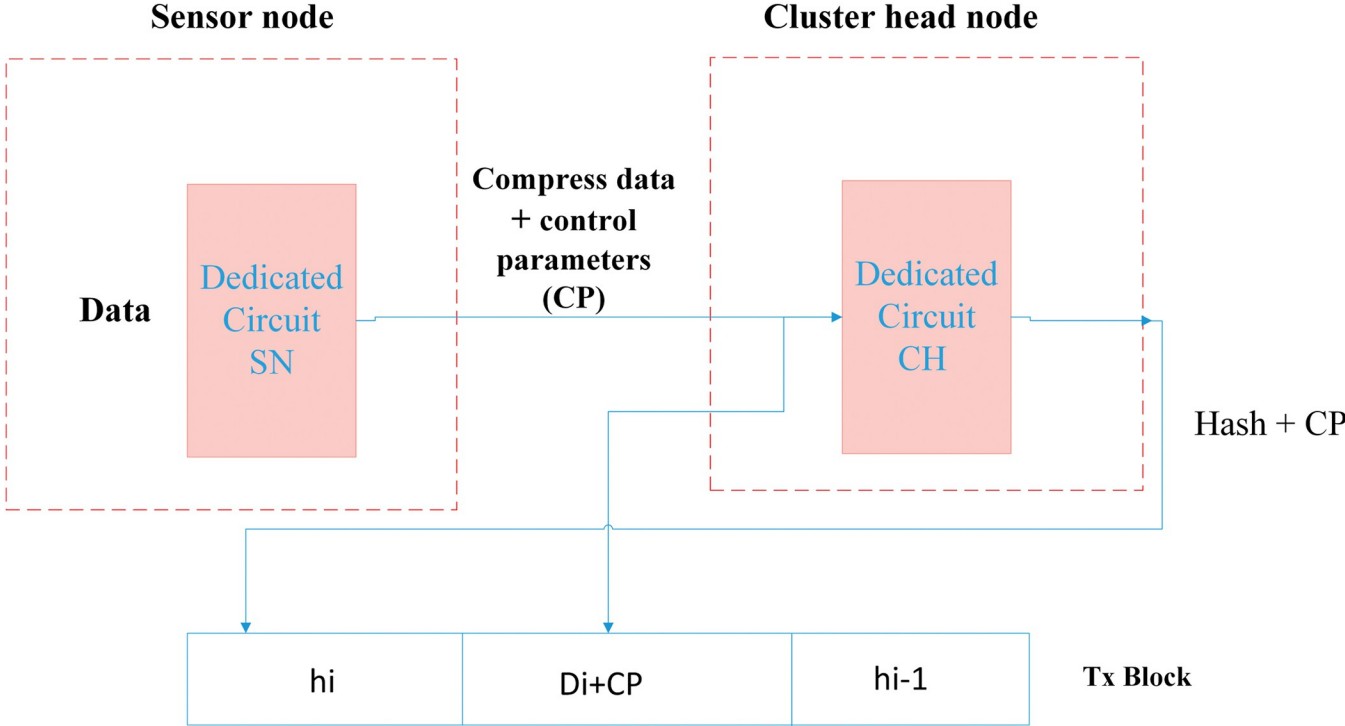

**Fig 4. Security system model: Hash creation and block generation.**

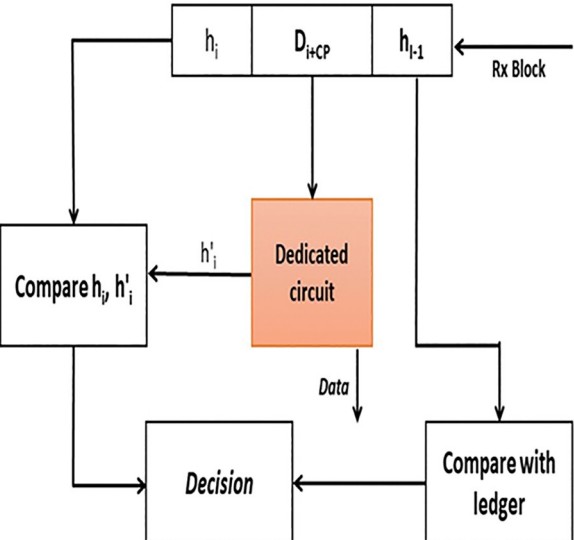

**Fig 5. Security system model: Authentication process.**

In the receiver, the hash value is generated again by entering the message and the control parameters that are explained in more detail in the next section. L, $\alpha$, and $X_0$ are sent within the block to the dedicated circuit to generate again the hash value as in [47], and the resulting hash value is compared with the current hash value. Hence, the parameters are entered into the dedicated circuit, the chaos sequence is generated again, and the hash is obtained to determine block authenticity. In addition, the hash value of the previous block, which was included in the current block, enables the other nodes to determine the source authenticity by comparing the 'previous' hash with their own ledgers. If both tests are passed, then the node will vote to accept the current block. If a node accepts a certain block after the authenticity process mentioned above, it will add it the chain stored in its ledger. Therefore, it will be considered as a 'previous' block in the next round. Once accepted by the block, it will be added to the blockchain in the ledger of each node, as shown in Fig 5.

## Circuit design and implementation

In this section, the two main function of the dedicated circuit are presented: the signal compression and the hardware-implemented algorithms for encryption and hash generation.

## Signal compression

The first function of the designed circuit is to compress the data based on the chaos theory for sampling. CS requires random sampling. Usually, uniform, Gaussian, or Bernoulli distributions are used. However, generating these random sensing matrices is complicated and requires high memory storage [48]. An alternative and less complicated way to generate these sensing matrices is to use chaos, which is generated using a deterministic approach with a few parameters that can be controlled by the sending and receiving parties. Reduced complexity is essential in WSNs with limited resources. In addition, the use of chaos introduces a level of security due to the sensitivity of chaos to its parameters. The control parameters consist of three values: length of the chaotic sequence L, $\alpha$, and the initial value $X_0$. They control the determination of the outputs of the chaotic system by certain initial states of these parameters. These chaotic equations are created through a simple chaos system, comprising the logistical

map determined by Eq (1).

$$X_{n+1} = aX_n(1 - X_n) \tag{1}$$

The sampling rate is reduced based on the chaotic sequence of the original signal by matching the sequence numbers resulting from the chaotic map and the signal indicator to determine which samples should be sent as a compressed signal according to the following Algorithm 1:

**Algorithm 1. Signal compression using chaotic logistic map.**
```
Input: Input L, α, b, d, sos, v[], x[0]
Output: Y (compressed data)
 1: for i = 1: L-1
 2: x[i] ← α x[i-1] *(1-x[i-1])
 3: end for
 4: for i = 0: L - 1
 5: x[i] ← b* (x[i]+d)
 6: end for
 7: i = 0
 8: while (i<sos)
 9: for j = 0: L - 1
10: i ← i *x[j]
11: if (i< = sos)
12: Y← v[i]
13: end if
14: end for
15: end while
```
Here, n = 0,1,. . . . . . . ., L-1.

- $X_0$ is initial value of chaotic logistic map.

- L represents the length of the chaotic sequence,

- α is a control parameter defined as a value to determine the appearance and behavior of the logistic map [49].

- d and b are constant set to the maximum required sampling interval [26]

- SoS represents the length of the original signal,

- V is the input signal,

- Y represents the value of the resulting compressed data.

To reconstruct the signal, the transmitting party must send the parameters L, α, and initial value $X_0$. By inserting these parameters into Eq (1), the same chaotic sequence as the original signal is generated.

As seen in Algorithm 2, linear interpolation is used to rebuild the compressed signal and return it to its original state. Moreover, as demonstrated in [26], interpolation is the act of predicting the unknown values that lie between the known values. A function's value can be roughly estimated at any location within its domain by using interpolation [50]. Interpolation is based on the assumption that the data points are connected by a smooth curve or surface. The value of the function at the target point is then calculated by fitting a curve or surface through the data points to obtain the interpolated value. In order to estimate the value of every point along a line that connects two known points, linear interpolation employs straight lines [51,52].

**Algorithm 2. Signal decompression.**
```
Input: Input L, α, b, d, sos, Y, x[0]
Output: V (original data)
```

```
 1: for i = 1: L-1
 2: x[i] ← α x[i-1] * (1-x[i-1])
 3: end for
 4: for i = 0: L - 1
 5: x[i] ← b* (x[i]+d)
 6: end for
 7: i = 0
 8: while (i<sos)
 9: for j = 0: L - 1
10: i ← i *x[j]
11: if (i< = sos)
12: id← i
13: end if
14: end for
15: end while
 16: V← interp {id,Y}
```

## Encryption and hash value generation

As stated before, in order to increase data security, the dedicated circuit will perform compressive sensing. The chaotic compressive sensing is designed to achieve simultaneous compression and encryption [32,53]. This encryption has many important features: security analysis showed that this resists brute force attacks, is sensitive to secret keys, robust against statistical attacks, and can use the parameters of the chaotic system as the secret key in the construction of the measurement matrix and also the masking matrix [32]. The computational complexity of these operations is much lower than that of the current mainstream RSA encryption scheme [54]. Compressive sensing can provide secrecy if the sensing matrix is changed for every measurement. This type of cryptosystem can be compared with One Time Pad since the sensing matrix is used one time [24].

In addition, the circuit is also responsible for generating the hash value that represents the backbone of blockchain technology and is also regarded as the main block identifier in the block chain. This method is proposed because it is characterized of being its low complexity. Theoretical analysis indicates that the algorithm can satisfy all the hash function requirements efficiently and flexibly. This method has higher sensitivity to the original message, which means that any change in the plaintext affects the ciphertext [47,55]. The hash is outlined in the following Algorithm 3.

**Algorithm 3: Generation of hash value [47].**
```
Input: L, α = 4, b, d, sos, v [], x [0],M (message)
Output: f(x) hash value
Hash generation
1: for i = 1: L-1
2: x[i] ← α x[i-1] * (1-x[i-1])
3: end for
4: Divide M into blocks
5: padding message M
6: for j = 0 to n-block
7: f(x) = M_i XOR x[i]
8: end
9: display f(x)
```

Thus, when cluster-head receives compressed and encoded data from Sensor node, the designed circuit creates a chaotic sequence, computes the hash and then composes the Tx block by encrypting the data with the previous hash.

Previous Algorithms 1–3 where coded in Verilog [56] language and synthetized into a FPGA. The ISE Design Suite 14.7 environment was used on a commercially available hardware

FPGA device (Family board: Artix7) [57]. From the FPGA implementation of the proposed circuit, power consumption results have obtained, as shown in next section. Hash generation and compressed sensing algorithms were also implemented in software to compare power consumption improvements with those of hardware.

## Experimentation and results

In this section the performance of our proposal, as well as its energy efficiency evaluation is analyzed. To do so, we will compare our proposal with other versions of WSN with blockchain without the dedicated circuit. The simulation and evaluation of the proposed method are accomplished in MATLAB version 2019b environment using the results obtained from the circuit implementation in the FPGA and software energy consumption.

## Compression signal results

In this section of our work, we'll examine the effectiveness of the signal compression algorithm, which works to reduce the size of data by sampling based on a logistic map. We'll also study the signal compression ratio we obtained using this approach and the statistical analysis of the signal compression ratio. In Fig 6 (1,2,3), a sample signal D(x) was generated and inserted into the dedicated circuit:

$$D(x) = \begin{cases} \dfrac{\sin(Nx/2)}{N\sin(x/2)} & x \neq 2\pi k, \qquad k = 0, \pm1, \pm2, \pm3, \ldots \ldots \\ (-1)^{k(N-1)} & x = 2\pi k, \qquad k = 0, \pm1, \pm2, \pm3, \ldots \ldots \end{cases}$$

**Fig 6.  (1,2,3) The original signals and compressed signals.**

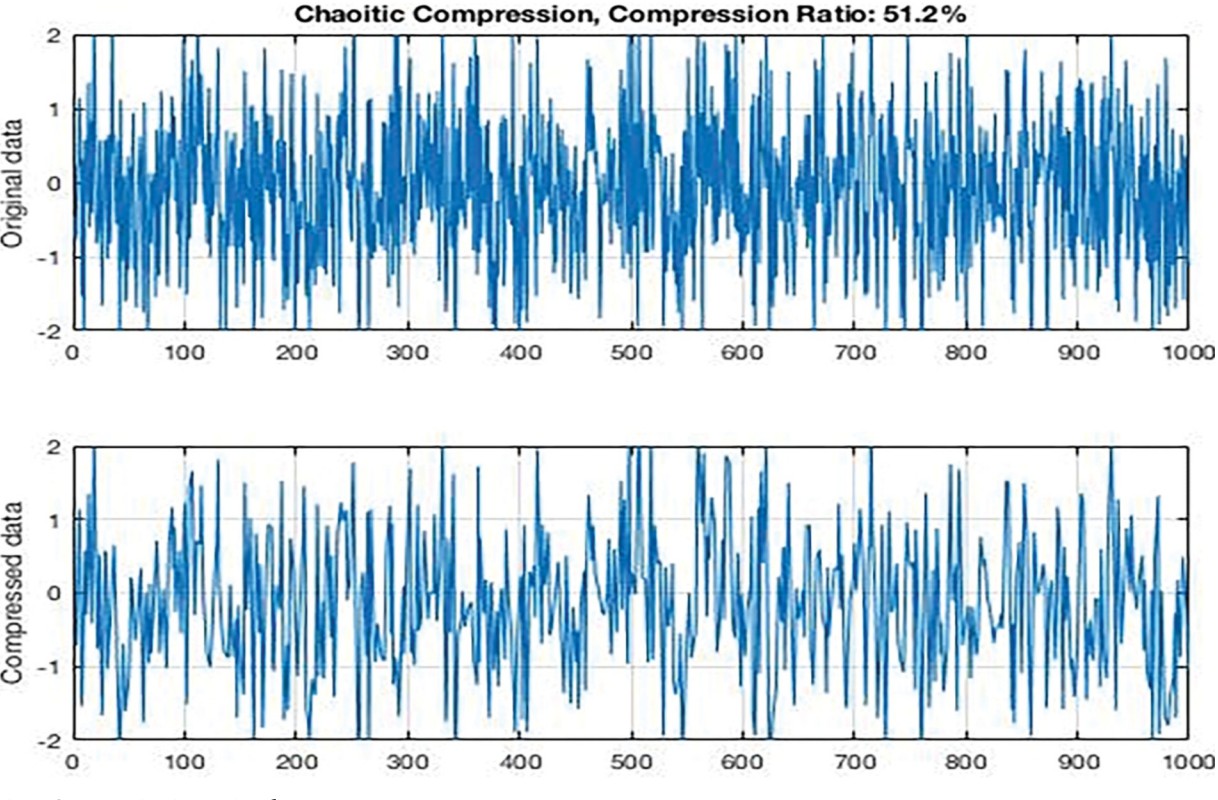

**Fig 7. Compression input signal.**

N integer value and nonzero When N = 5, 4 and 3.

In Fig 7 signal G(x) was generated as a random digital signal, determinate, and discrete and inserted into the dedicated circuit.

In the compressed sensor approach, linear compression approach is used to transform a signal X original data into another form Y compressed data. Resulting in compressed signals, the length of the signal time period was random between 500 and 1000 and the number of signal points was 1000.

The compression ratio data has been obtained by the statistical SPSS [58] and the results are shown in Fig 8 and Table 2. We have noted from the data analysis that the mean of the data is 51.167%, the median is 51.174%, and the coefficient of kurtosis is 0.355. This indicates that the data is centered on its mean, and approaches the normal data distribution. We also have noted that the standard deviation is 0.077, the variance is 0.006, and the lowest value is 50.87 while the largest value is 51.30. Therefore, the range is 0.431, and the deviation is 0.582. This indicates a high degree of data convergence and low dispersion.

These results about compression ratio were introduced in the MATLAB environment to simulate the circuit operation together with the next ones obtained with the FPGA and software execution.

## FPGA results of the dedicated circuit

After developing, synthesizing and checking the correct operation of the algorithms described in the previous sections in a FPGA, we obtained its power consumption results. The VIVADO ® design suite was used for this purpose. The power consumption of the implemented

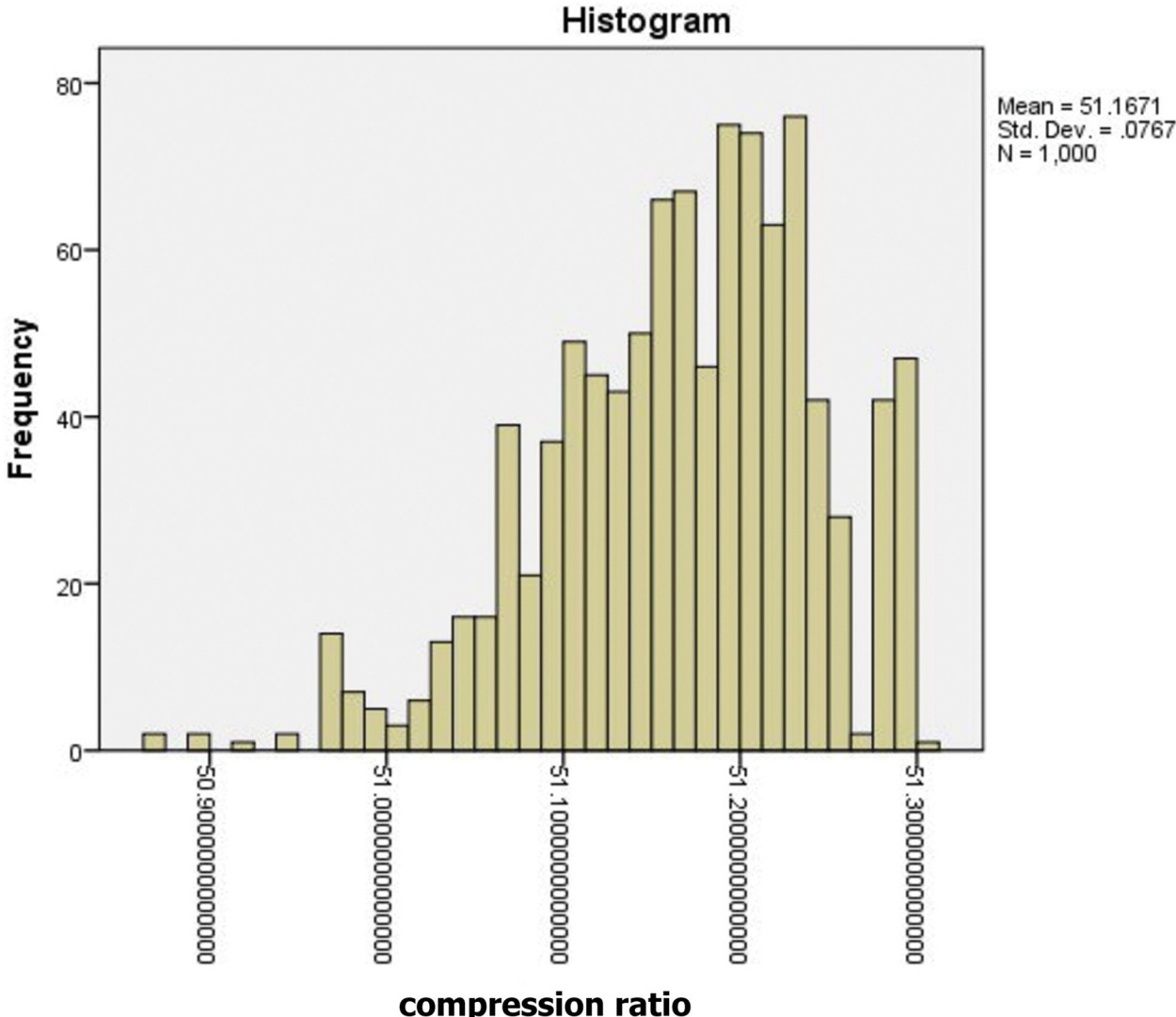

**Fig 8. Descriptive statistics of signal compression.**

dedicated circuit for compression is shown in Table 3. It can be noted that the consumption is about 0.082 W when the signal size is 752 bits.

On the other hand, the consumption power of the implemented hash generation circuit is 0.663 W as shown in the Table 3.

To compare with software-implemented functions, hash SHA256 generation algorithm was implemented in an ultra-low power microcontroller. A STM32L476) ST. STM32L4 MCU series Ultra-low power with performance (was selected as a representative microcontroller of actual embedded applications. Data from STM 1shows a power consumption equal to 28 μA/Mhz using an external switched-mode power supply (SMPS) and operating at 24 Mhz. This is the minimum consumption in an operational stage. With this consumption per Mhz, the best results (minimum consumption) can be obtained. Additionally, results were also obtained for a more common case. An experiment was performed with the microcontroller running at 80 Mhz (its maximum frequency) and without external switched-mode power supply. This could be representative of many actual designs.

**Table 2. Statistics descriptive.**

| | | Statistic | Std. Error |
|---|---|---|---|
| Mean | | 51.1671 | .0024 |
| 95% Confidence Interval for Mean | Lower Bound | 51.1624 | |
| | Upper Bound | 51.1719 | |
| 5% Trimmed Mean | | 51.1703 | |
| Median | | 51.1744 | |
| Variance | | .006 | |
| Std. Deviation | | .07672 | |
| Minimum | | 50.8704 | |
| Maximum | | 51.3011 | |
| Range | | .4307 | |
| Interquartile Range | | .1046 | |
| Skewness | | -.582- | .077 |
| Kurtosis | | .355 | .155 |

In ("Design recommendations for STM32L4xxxx with external SMPS, for ultra-low-power applications with high performance") different consumption benchmarks results are shown. As stated in that application note2 for an 80 Mhz frequency, a consumption about 120 μA/ Mhz represents a real situation. As there is a linear relationship between the software function execution time and the energy, the execution time of those algorithms were computed and then the consumption of these algorithms was obtained. Thus, we will also compare the system with these functions implemented in software and the proposal with functions implemented in hardware to demonstrate the advantages of the latter. Therefore, comparing software and hardware depends on adding/eliminating the complexity of the computation time and its associated power consumption. That is, when functions are not carried out by hardware, software complexity will be higher. Table 4 shows the consumption of the software-implemented algorithms:

Additionally, to study all the possibilities, a software compression technique was also designed in the software-only version to analyze its benefits. Then, a software version of the Chaos compression technique implemented in hardware was implemented on the same microcontroller. Therefore, a comparison between software with and without compression techniques has been carried out, in addition to comparing them with the hardware design will be conducted.

## Results

This section presents the results obtained with the proposal to optimize the power consumption of wireless sensor nodes. Our hardware-implemented solution will be compared to the standard implementation of a wireless sensor network (WSN) with blockchain technology for securing the system. To obtain the results, MATLAB tool will be used to configure and simulate the WSN described in section 3. It use the well-known Leach algorithm [34,35] to coordinate and maintain the network block structure and determine routing strategies. To perform

**Table 3. Hardware dedicated circuit power consumption in CS and hash.**

| Function of Dedicated circuit | Power consumption / watt | signal size |
|---|---|---|
| CS | 0.082 W | 752 bits |
| Hash | 0.663 W | 752 bits |

**Table 4. Power consumption for SHA256 software algorithm.**

|  | 24 Mhz with SMPS | | 80 Mhz without SMPS | |
|---|---|---|---|---|
| Algorithm | Execution time | Consumption | Execution time | Consumption |
| SHA algorithm *(3800 bits of input data)* | 120 ms | 0,0002611 J | 36 ms | 0,00114998 J |

the MATLAB WSN model, the results obtained from the synthesis and implementation of the dedicated circuit using the FPGA will be introduced to MATLAB WSN model. After that, power consumption will be obtained from MATLAB simulations. Our proposal will be compared as follows:

1. We will obtain the power consumption and network lifetime for a WSN with Leach algorithm and a traditional software-implemented blockchain (i.e no data compression and hash computation obtained by software).

2. We will compare the previous results with a WSN with Leach algorithm and a software implemented blockchain (i.e without the dedicated circuit) but with chaos compression algorithm. The compression technique (chaos) must be responsible for part of the improvements in the network lifetime. Thus, we decided to measure the benefits and drawbacks (extra consumption) of implementing the chaos compression in the software.

3. Finally, we will obtain the results of our proposal: a WSN with Leach algorithm and a hardware implemented support (dedicated circuit) for blockchain operations.

In order to obtain the energy consumption of the nodes, the next steps will be taken into account:

- Each sensor node reads input data and compresses data. Then it sends the data to is cluster-head.

- The cluster-head collects data from sensor nodes and generates a hash to form a block.

- The node's energy is re-calculated by subtracting the transmitter energy consumption from the initial energy, so that the new node energy is [34].

$$E'_{\text{new}} = \begin{cases} E_{\text{old}} - \left(E_{DA} + E_1 \times N_{\text{bits}} + E_{fs} \times N_{\text{bits}} \times d^2\right) if \ d < \sqrt{\dfrac{E_{fs}}{E_{\text{amp}}}} \\ E_{\text{old}} - \left(E_{DA} + E_1 \times N_{\text{bits}} + E_{\text{amp}} \times N_{\text{bits}} \times d^4\right) if \ d \geq \sqrt{\dfrac{E_{\text{amp}}}{E_{\text{amp 2}}}} \end{cases} \tag{2}$$

Where $E_{DA}$ is the data sensorization energy, and it is only consumed on the transmitter node, as it collects data using its input peripherals. $E_1$ is the energy consumed by the transmitting and receiving operations and depends on factors such as the signal coding, filtering or

**Table 5. Power consumption for software compression–chaos.**

|  | 24 Mhz with SMPS | | 80 Mhz without SMPS | |
|---|---|---|---|---|
| Algorithm | Execution time | Consumption | Execution time | Consumption |
| CS—*1000 input data (x 16 bit)* | 2,445 ms | 5,42203E-06 J | 734 µs | 2,34469E-05 J |

**Table 6. Simulation environment parameters.**

| Parameters | Value |
|---|---|
| Network area size | $100 \times 100$ |
| Nodes number | 100 |
| Initial energy | 0.5 J |
| Maximum block size | 1000 bits |
| $E_{amp}$ (Energy of amplifier) | 0.0013 pJ/m$^2$ |
| $E_{elec}$ (energy consumed to transmit or receive) | 50 nJ/bit |
| $E_{fs}$ (amplification coefficient of signal) | 10 pJ/bit/m$^2$ |
| Dedicated circuit (compression signal) | 1.5530e-12 J/bit |
| Dedicated circuit (hash generation) | 2.32e-09 J/bit |
| Software consumption (for compression and SHA256 functions) | See Tables 4 and 5 |

modulation. $E_{fs}$ (free space) and $E_{amp}$ (multi-path) fading channel models depends upon the transmission distance d. If the distance is below a threshold, the free space model is used. In other case, the multi path model is chosen.

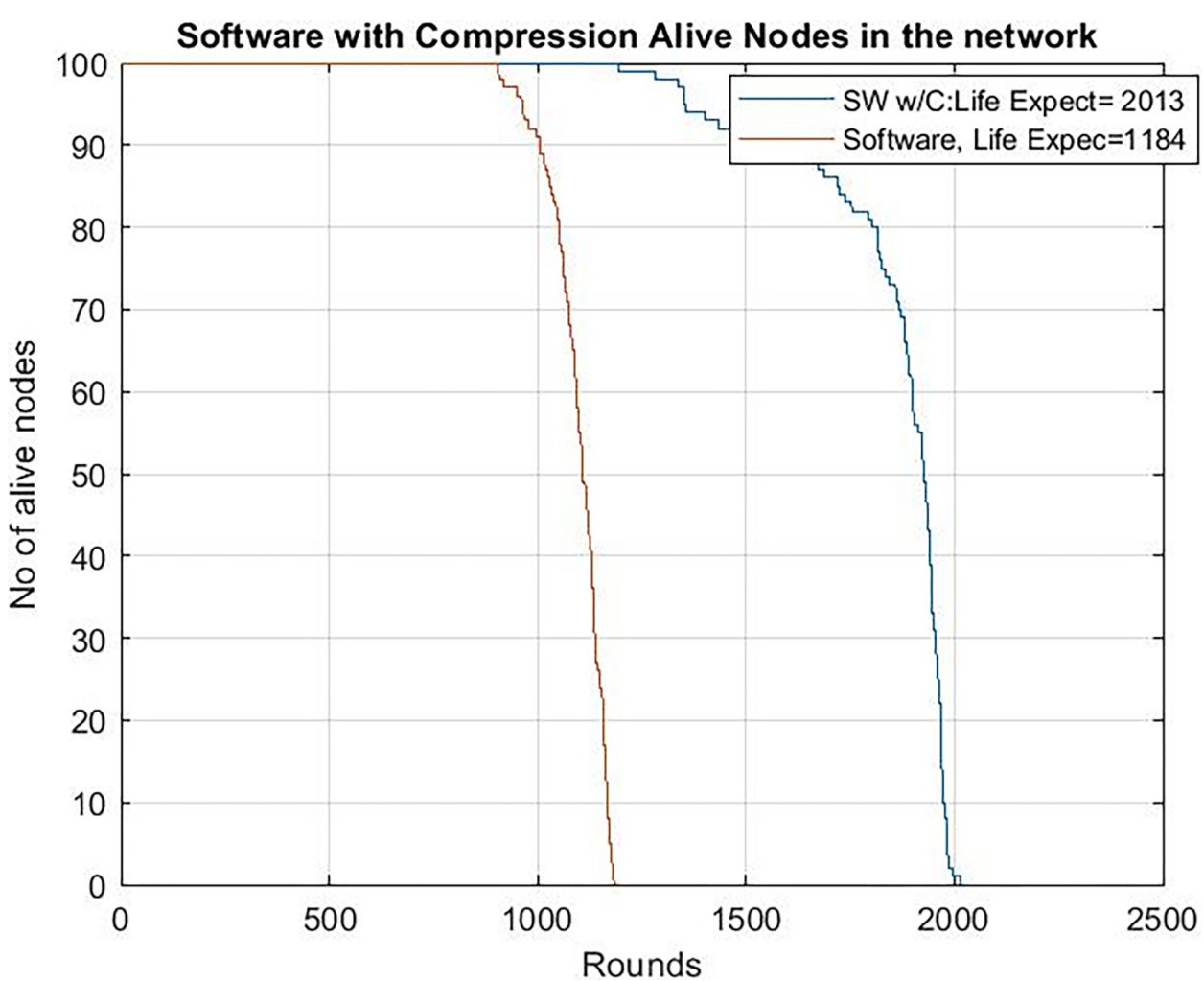

**Fig 9. Alive nodes clock frequency = 24 Mhz in the network with chaos compression as software and without software chaos compression.**

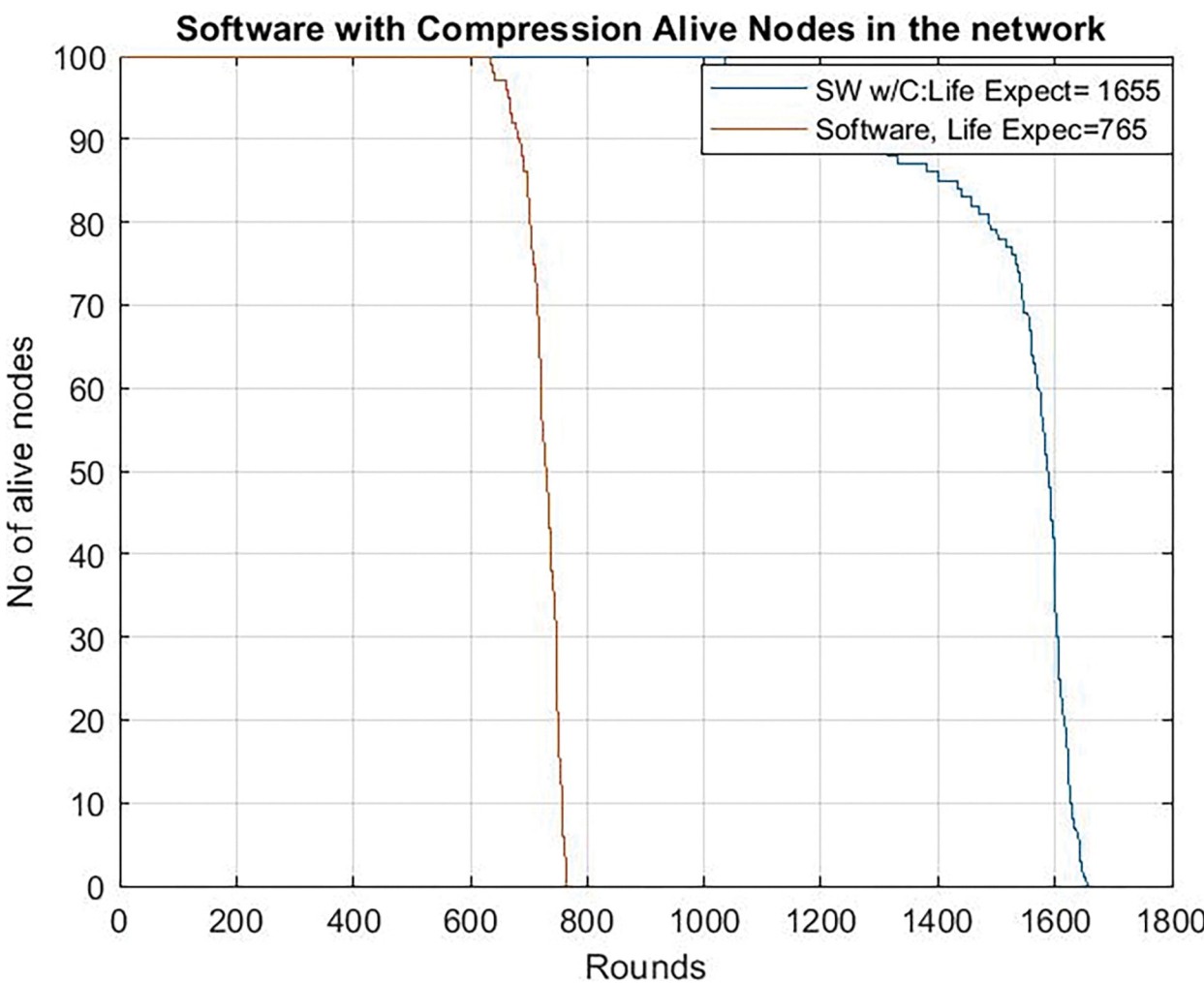

**Fig 10. Network lifetime.** Clock frequency = 80 Mhz chaos compression implemented software (SW w/C) version / software version without chaos.

In addition, the consumption of the dedicated circuit (obtained from the FPGA implementation) to fulfill the data compression and hash computation is added. In the "hardware" version, it is assumed that the node uses a separate hardware circuit to implement hash generation and compression, while basic utilities are performed by the microcontroller. The consumed energy at each node is then calculated as follows:

$$E_{new} = E'_{new} - E_{Hardware} \times N_{bits} \tag{3}$$

where $E_{new}$ is the energy of the node after deducting the basic utilities consumption as in Eq 2 above.

For comparison purposes, the hardware hash computation and chaos compression have also been implemented in software, and their energy consumption have obtained as reflected in Tables 1 and 2 for different microcontroller configurations. This way, for software implementations:

$$E_{new} = E'_{new} - E_{Software} \tag{4}$$

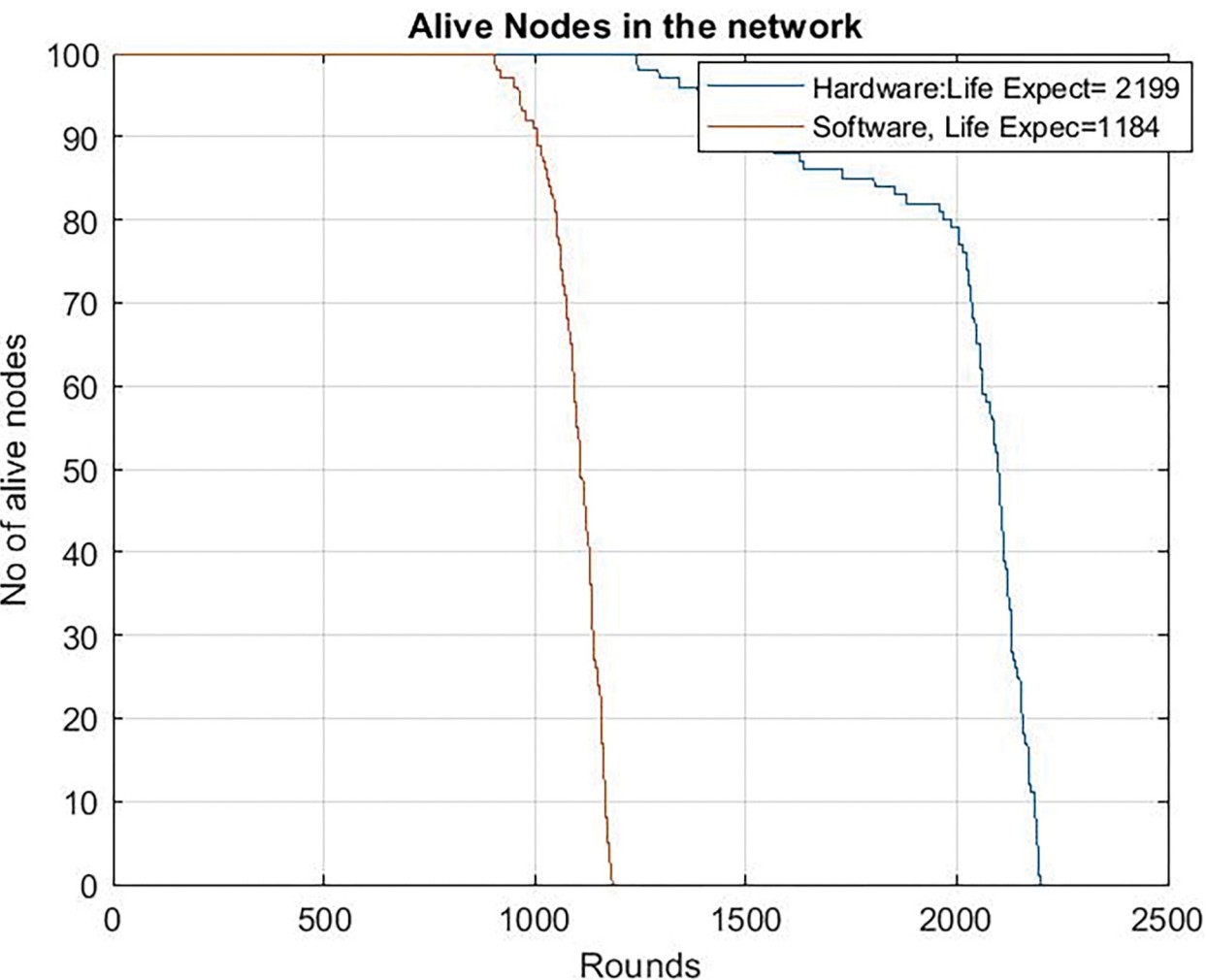

**Fig 11. Alive nodes in the network with hardware compression and with software tradition blockchain clock frequency = 24 Mhz.**

## MATLAB simulation scenarios

The MATLAB simulation scenarios used in the research have the following assumptions:

- Sensor nodes were distributed in a random way in a square 100 m×100 m region.

- Sensor nodes were characterized by being homogeneous. They are assigned a matchless identifier number in the network, and their energy is limited.

- The base station is located in the center of the square with a static position.

- The power of the wireless transmitter could be adjusted. Table 6 lists the basic parameters.

Several simulations were performed to analyze the performance improvement of the proposed solution compared with software approaches. This performance centers on two main parameters: Network lifetime and total energy consumption.

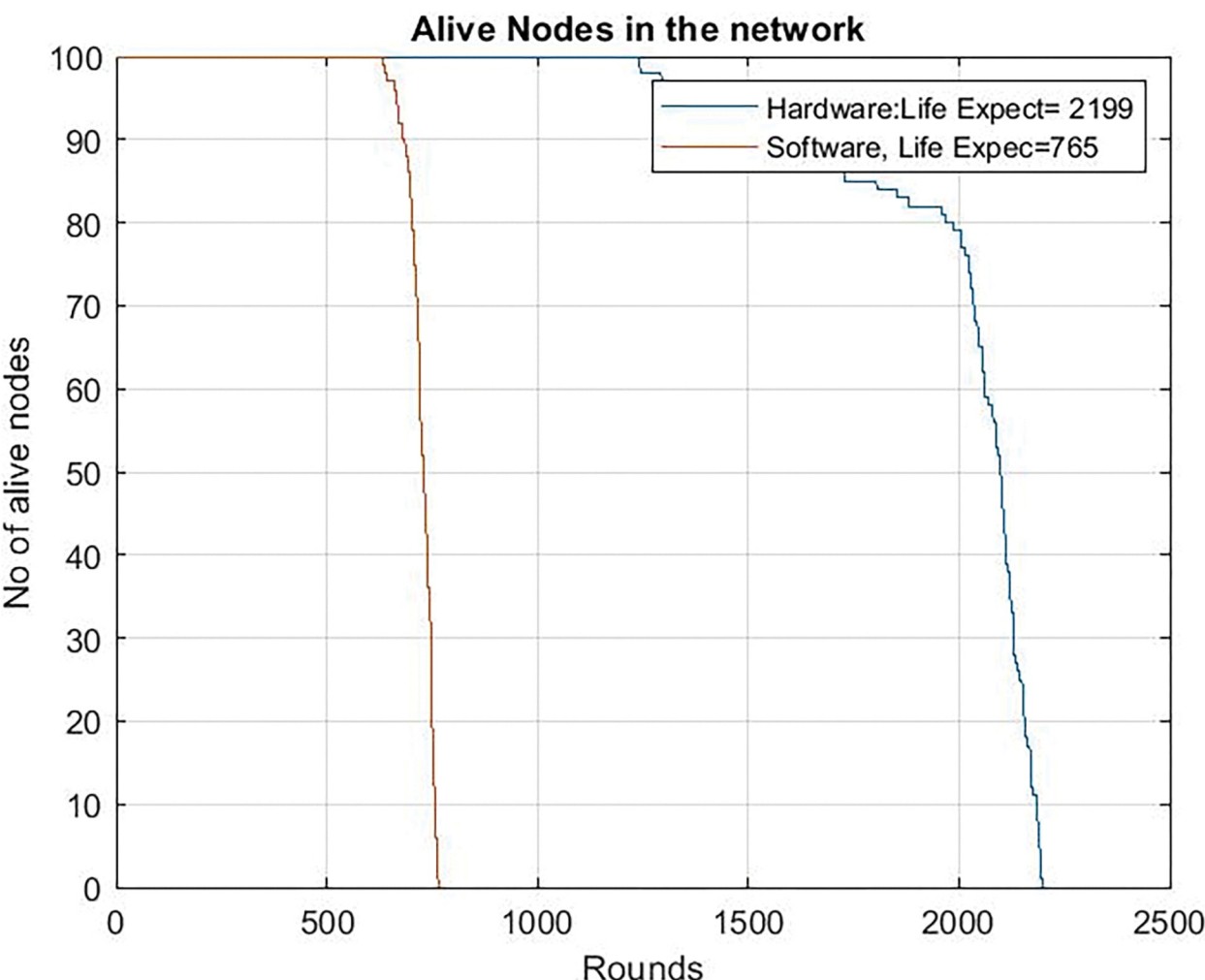

**Fig 12. Alive nodes in the network with hardware compression and with software tradition blockchain clock frequency = 80 Mhz.**

## Network lifetime

In this research, the definition of network lifetime can be described as the time from the start of the simulation to the time when the energy in the battery of the last node does not allow its operation (the node dies). In WSNs, the lifetime of the network can be classified into two periods, as the <u>stable period</u> and <u>unstable period</u>.

- Stable period typically refers to the lapse of time between the start of the simulation and the time when the first node dies.

- Unstable period refers to the time from stable period to the end of the simulation.

Fig 9 shows the network lifetime for the two first studied configurations. It illustrates the network lifetime of a WSN without our proposal (i.e a standard WSN with Leach protocol and a software-implemented blockchain without data compression and hash computed by the software) and a WSN with, in addition, software chaos compression (labelled SW w/C in the figure). Simulation results show that the network lifetime with the proposed enhanced software solution is greater than the lifetime without compression. When no data compression is performed and hash computation is done by the software, 100% of the nodes died at round 1184,

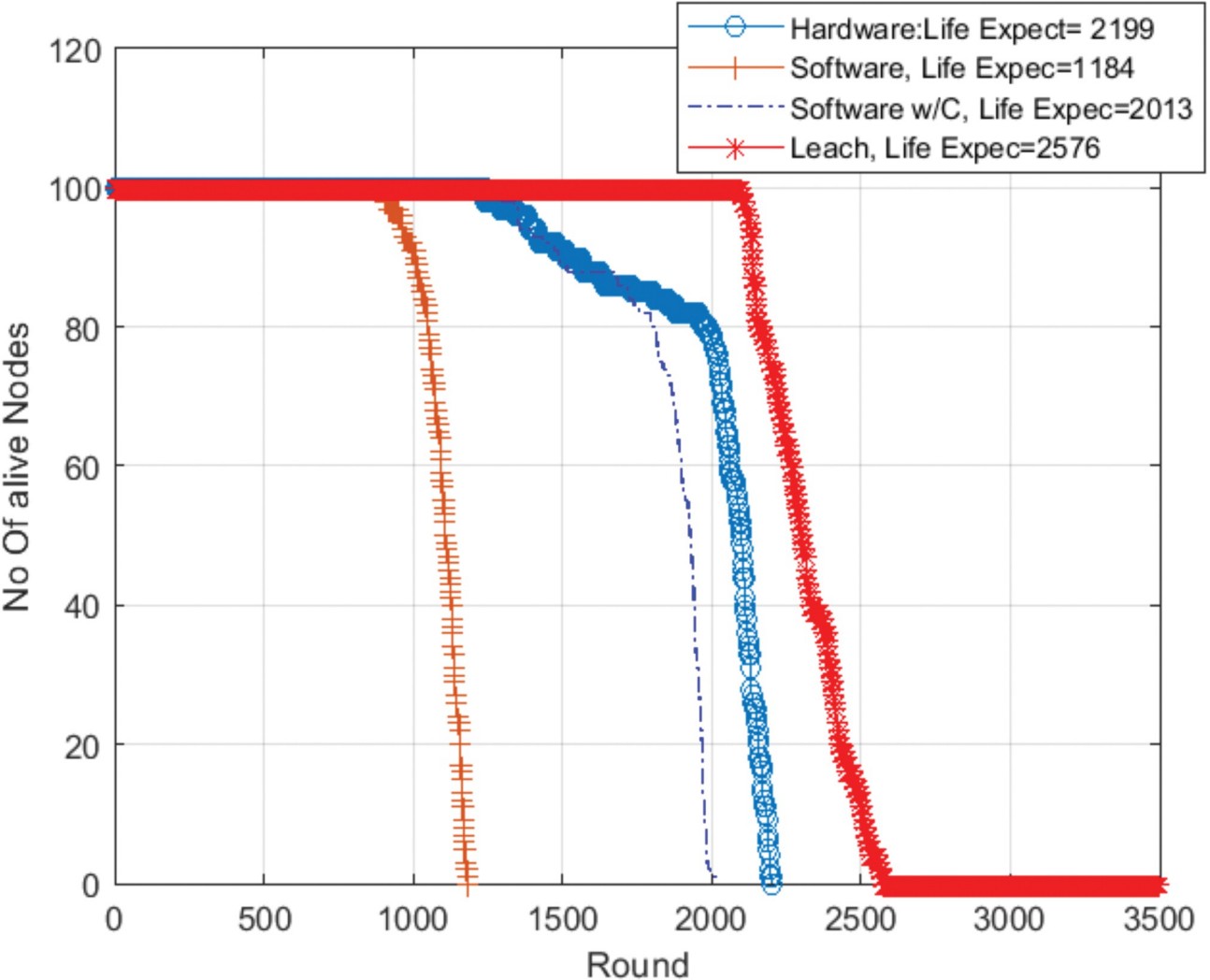

**Fig 13. Alive nodes in the network with (leach without BC, BC with hardware compression, and BC software tradition) clock frequency = 24 Mhz.**

while when chaos compression techniques are used (by software), the nodes died at round 2013.

So, chaos compression techniques, even in software version, significantly improves the network lifetime. The benefits of compression techniques may outweigh the extra consumption of the algorithm due to the less data to transmit. However, it remains to be seen what happened if the microcontroller works at a higher frequency (and then with higher consumption).

Fig 10 shows the results when the microcontroller works at 80 Mhz. In this case, with more power consumption, the use of software chaos compression techniques significantly increases the network lifetime (from 765 to 1655 rounds).

And now we are going to compare previous results with the dedicated circuit version. Fig 11 shows the benefits of doing the blockchain operations with the dedicated circuit, even when working in an ultra-low power mode in the microcontroller. The network lifetime is extended from 1184 to 2199 rounds.

On the other hand, Fig 12 shows the same results when the microcontroller works at 80 Mhz. In this case, as the consumption of the microcontroller is higher, only 765 rounds are

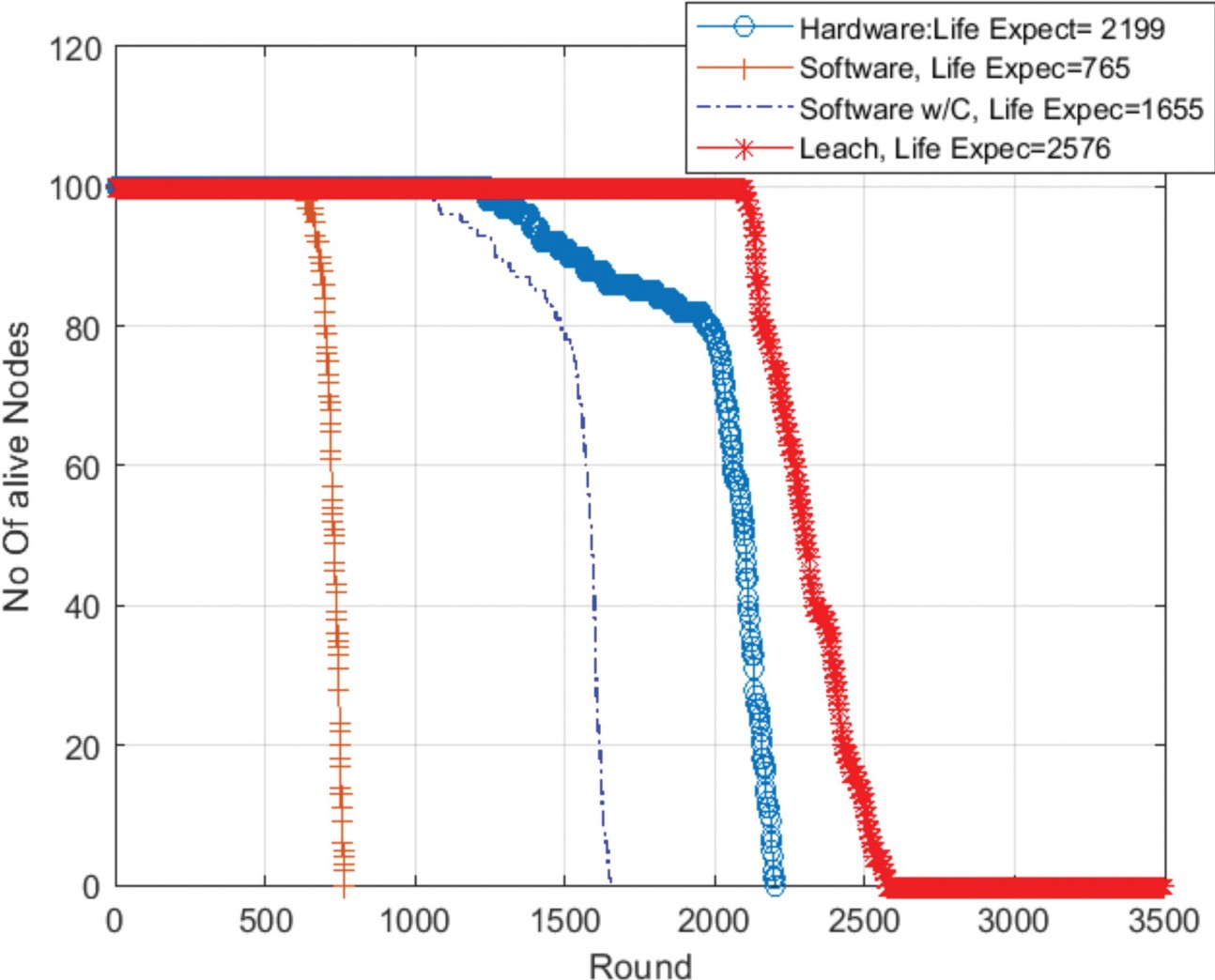

**Fig 14. Alive nodes in the network with (leach without BC, BC with hardware compression, BC with software compression, and BC software tradition) clock frequency = 80 Mhz.**

obtained for the software version. The use of the dedicated circuit for blockchain and compression functions increases by more than three times the lifetime of the network.

It should not be forgotten that the inclusion of blockchain techniques in WSN will provide great benefits at the cost of higher consumption. Minimizing this increase, in order to make this possibility viable, is the reason for this work. To show this concept, in Figs 13 and 14 we have added the network lifetime of a WSN that uses Leach protocol but without blockchain to compare with the previous results.

Leach protocol case, without blockchain, shown by the red color in Figs 13 and 14, exhibits the longest lifetime among other technologies. Fewer operations, less consumption and longer lifetime. But obviously, it does not benefit from the desired advantages of blockchain technology for securing the system.

When using a security technology such as a blockchain, energy consumption will increase, which will make the network lifetime shorter. Software implemented blockchain without data compression is represented in brown color. Moreover, when our chaos compression proposal was used as software, there was a clear improvement in the network lifetime as shown in the

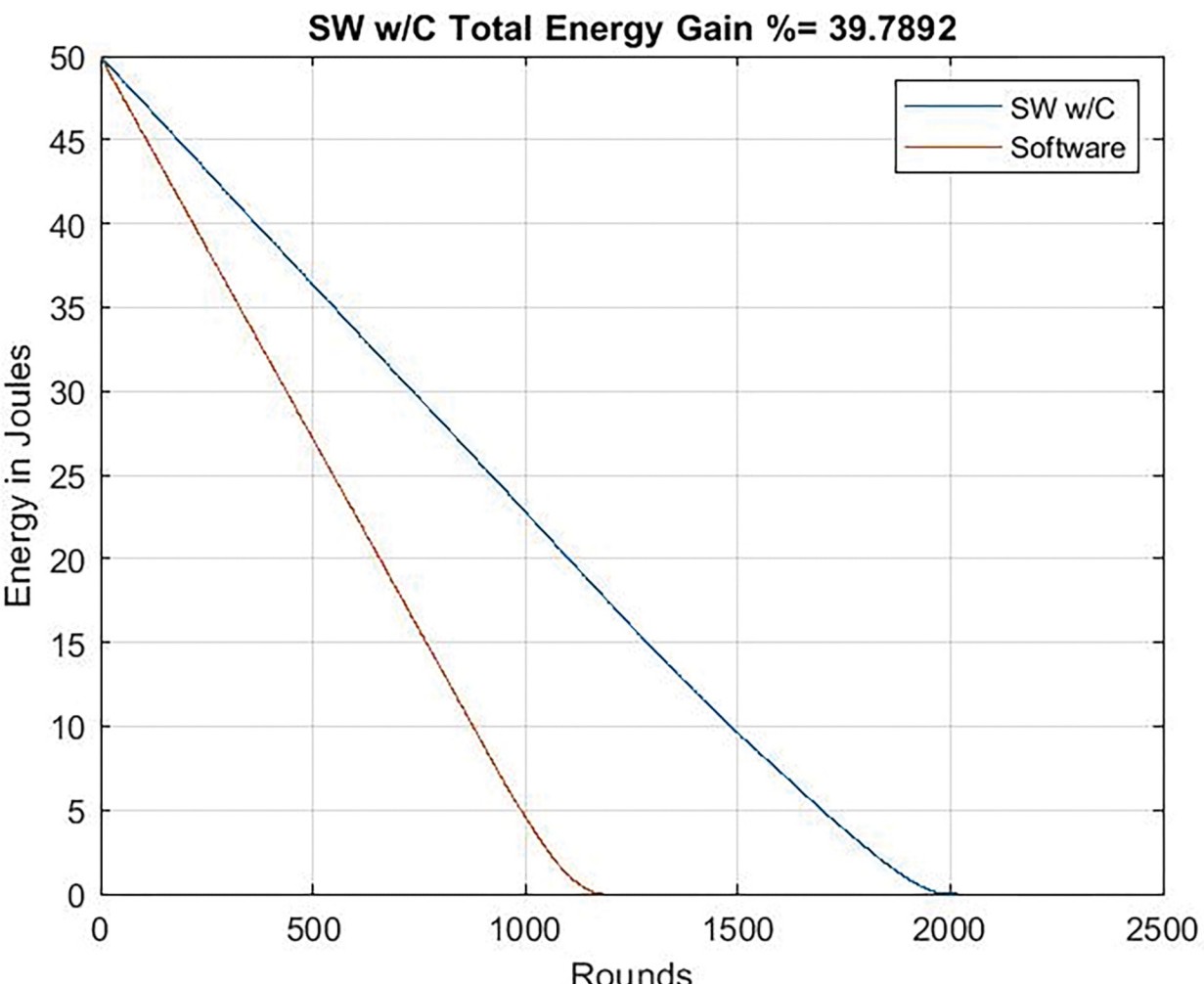

**Fig 15. Energy gain.** Clock frequency = 24 Mhz chaos compression implemented software (SW w/C) version / software version without chaos.

blue color. This improvement increased significantly when using our proposal as hardware compared with the traditional blockchain or the proposal as software as shown in the bold blue color. They are close to the of the non-blockchain version. Thus, the use of blockchain becomes a real possibility.

## Total energy consumption

As it can be seen in Figs 15 and 16, the reduction of data transmission due to compression allows an important energy saving: 39.7% when running at 24 Mhz and 52% when running at 80 Mhz. Obviously, results with the dedicated circuit are better, but these results highlight the importance of minimizing the workload to transmit by means of compression techniques.

Figs 17 and 18 illustrates the energy gain curve that refers to the difference between the energy expended (or conserved-depending of how you look at it-) in software and hardware versions. Even when the microcontroller works minimizing its consumption (24 Mhz), the dedicated circuit allows an energy gain equal to 43%. If the microcontroller works at a higher frequency, the benefits are clear: 63%, that it is to say, when the software-based system deaths, the system with the dedicated circuit still has the 63% of energy.

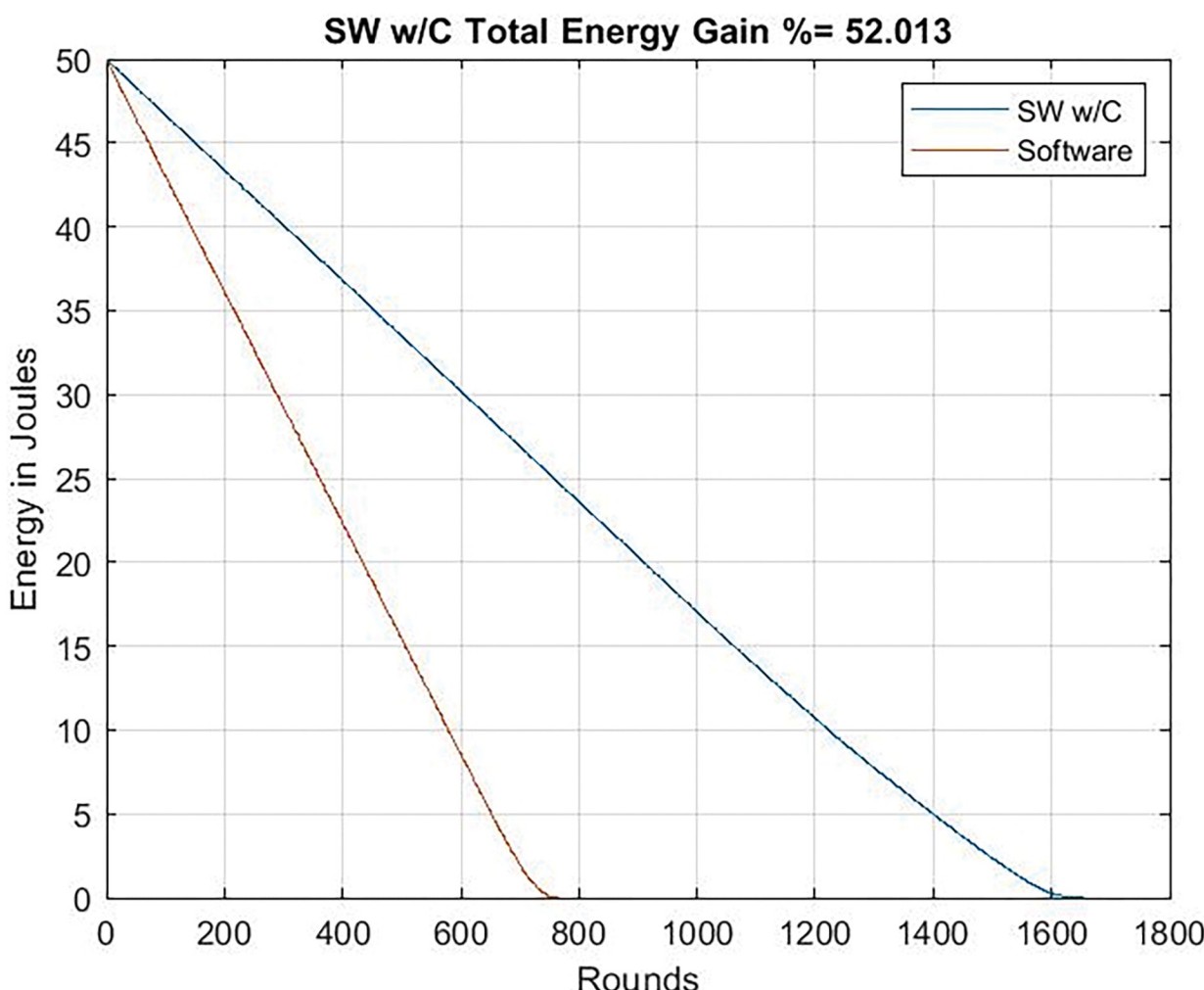

**Fig 16. Energy gain.** Clock frequency = 80 Mhz chaos compression implemented software (SW w/C) version / software version without chaos.

### Energy variance

Regarding the energy variance of the nodes, it was very high in the traditional software approach, while when using the software compression, it decreased as shown in Fig 19 in the case of clock frequency = 24 Mhz, as well in case of Clock frequency = 80 Mhz as shown in Fig 20.

When the dedicated circuit is replaced by software to compress the data, it is also shown that the energy variance of the nodes was very high. When using the dedicated circuit, the energy variance decreased as shown in Fig 21, in case of Clock frequency = 24 Mhz and, also in case of Clock frequency = 80 Mhz as shown in Fig 22.

The hardware solution enables the WSN to live longer than the traditional one. Therefore, the power variance of a WSN containing a dedicated circuit continues to rise during the rounds while the conventional network loses power.

### Conclusions and future work

Adopting a blockchain in a WSN is not easy. It is necessary that several critical limitations (energy, memory size, computation load and power consumption) are addressed. This

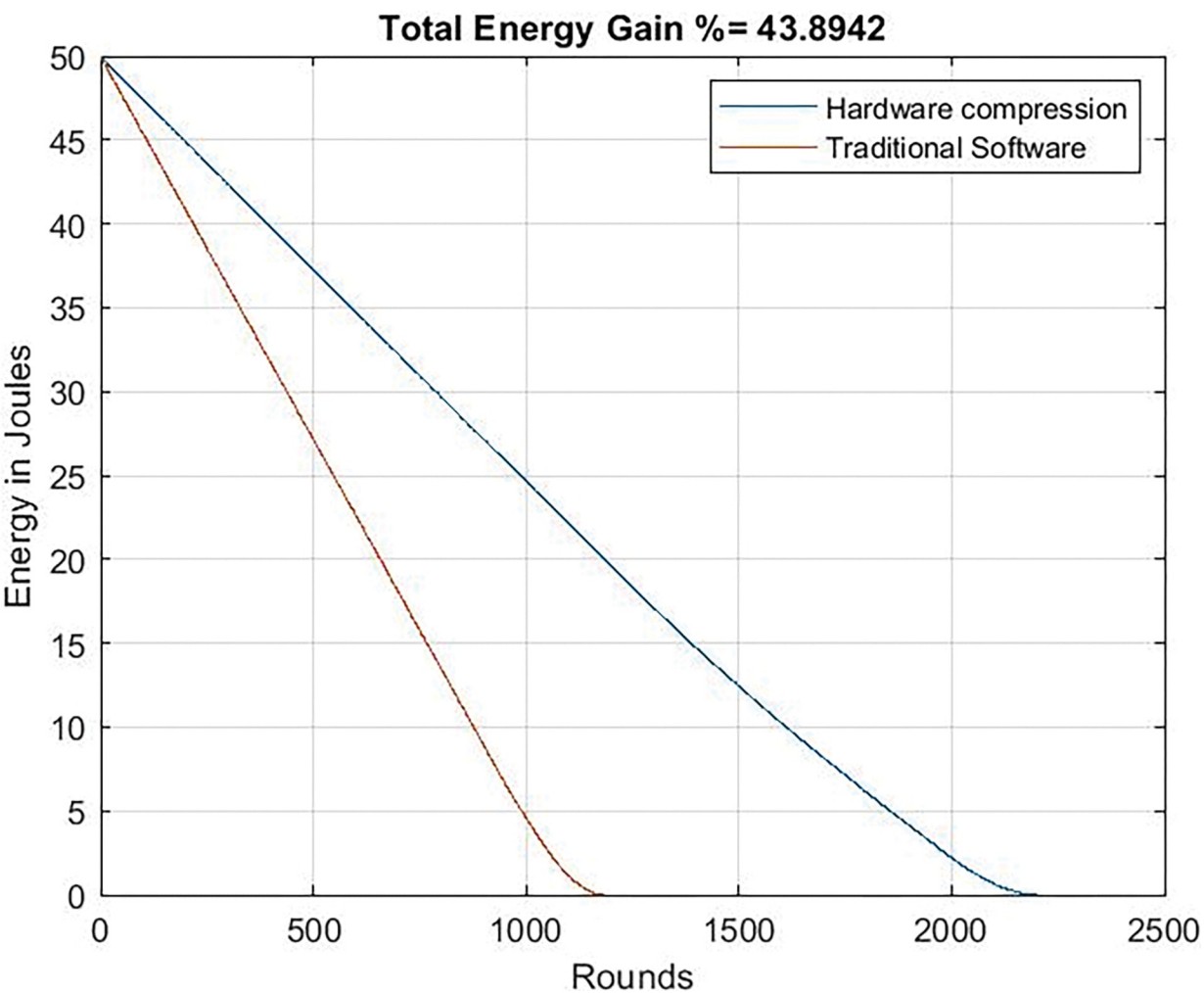

**Fig 17. Energy gain.** Clock frequency = 24 Mhz.

proposal contains a step forward in solving these main limitations for adopting blockchain technology in WSN. To overcome these problems, a dedicated circuit is designed to perform some blockchain-related functions, such as hash generation and data compression. This paper has demonstrated how the hardware implementation of these complex software functions contributes significantly to reduce the consumption of energy of the entire network and extend its the lifetime.

This way, hardware signal compression is an effective method to reduce the data to transmit and, therefore, to minimize the energy requirements. Thus, transferring the software functions (hash computation) and compression techniques to a specific circuit reduce both the time and the energy requirements to compute them. The MATLAB platform was used to test the new proposed approach. In order to provide the real operation parameters to the MATLAB simulator, a Verilog implementation of the dedicated circuit was performed. From this real implementation, energy consumption was obtained. The energy consumption of the software functions performed by the dedicated circuit was also obtained by implementing them in a representative embedded microcontroller. Real energy consumption was measured in laboratory conditions and results were added to the MATLAB simulations. The main results show a

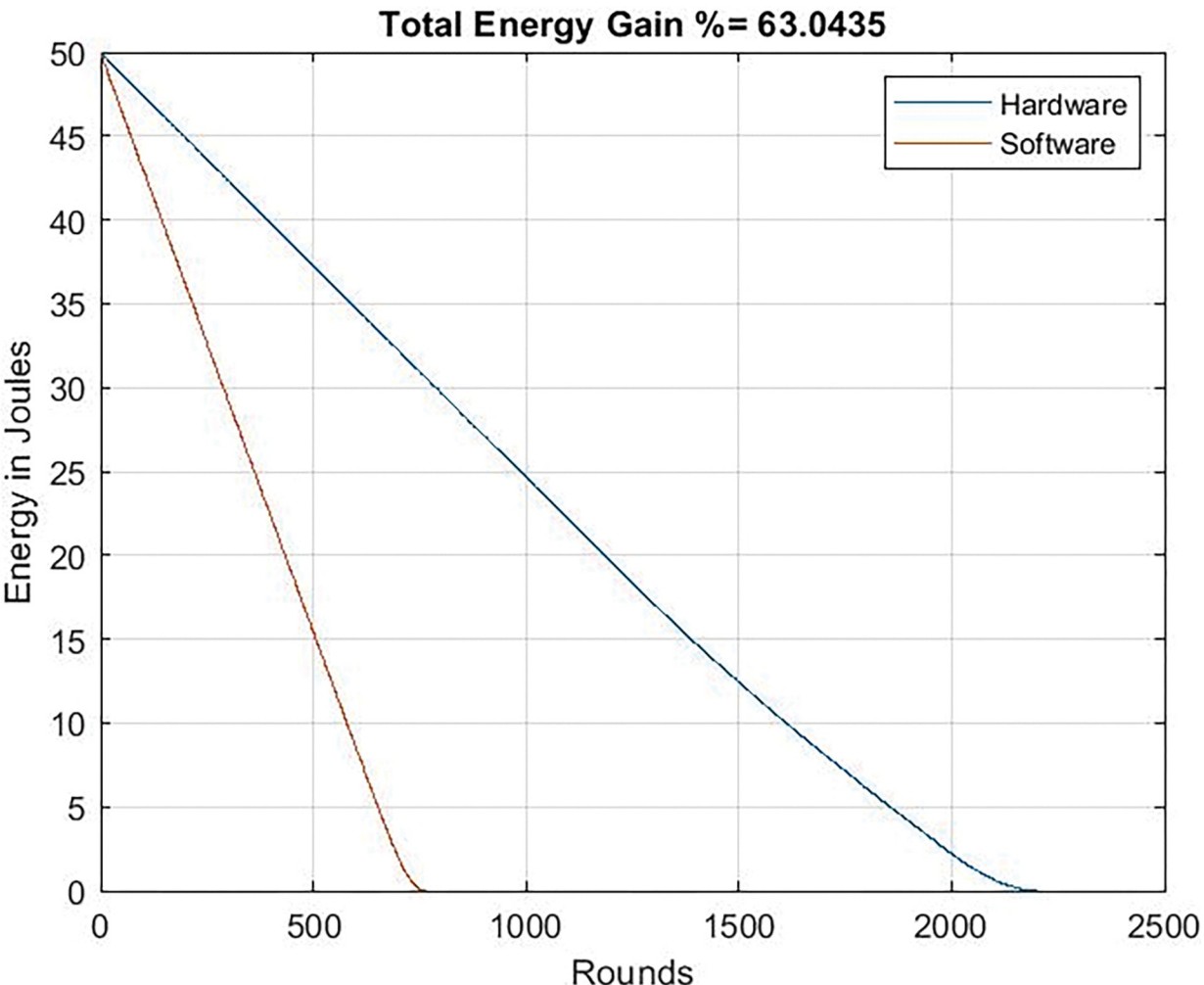

**Fig 18. Energy gain.** Clock frequency = 80 Mhz.

reduction of up to 63%. Therefore, the feasibility of implementing blockchain in WSN with dedicated circuits has been demonstrated. This opens the door to new implementations and uses of the blockchain technology in WSN.

Compared to the systems presented in the States of the Art section, our proposal offers several advantages. This way, regarding [35], where the researchers use public and the private key with a high energy consumption associated with the encryption and decryption processes and a transaction validation process performed in the master node, our approach solved this challenge by proposing a dedicated electrical circuit for the purpose of compressing and encrypting data simultaneously to reduce complexity and energy consumption. As for the choice of cluster head, it is conducted by the remaining energy, a solution presented by the researchers to improve energy consumption. Our proposal adds a simple and straightforward data compression process to reduce the volume of transmitted and received data to improve energy consumption. Regarding research [36] in which different traffic flows try to reduce the energy loss, our work is based on data compression as a way to reduce energy consumption. Moreover in [36] the reward mechanism is adopted after each round while in our approach we use precedence in the process of sending to base station. On the other hand, our proposal improves [37]

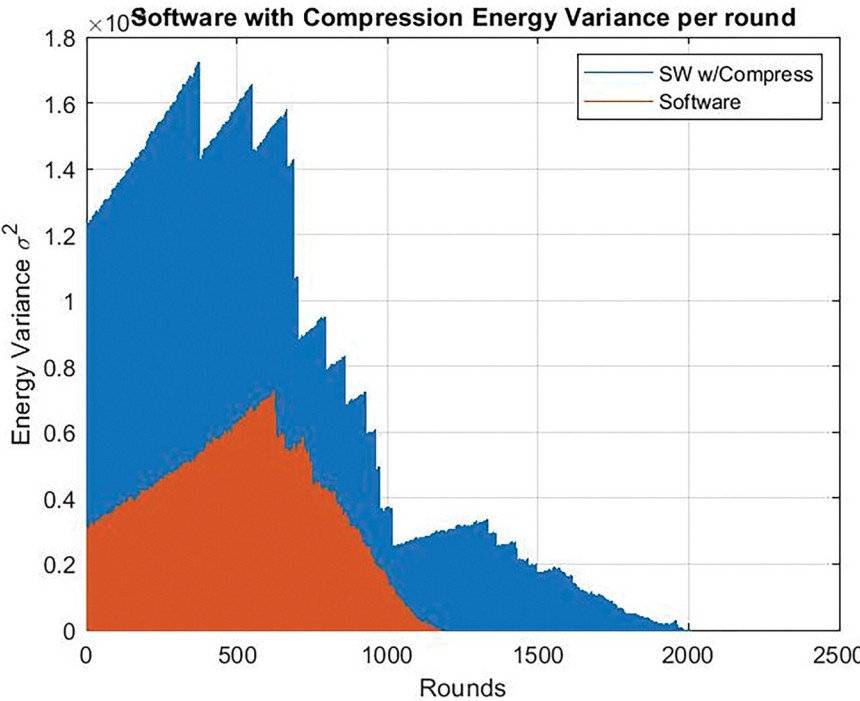

**Fig 19. Energy variance per round.** Clock frequency = 24 Mhz chaos compression implemented software (SW w/C) version / software version without chaos.

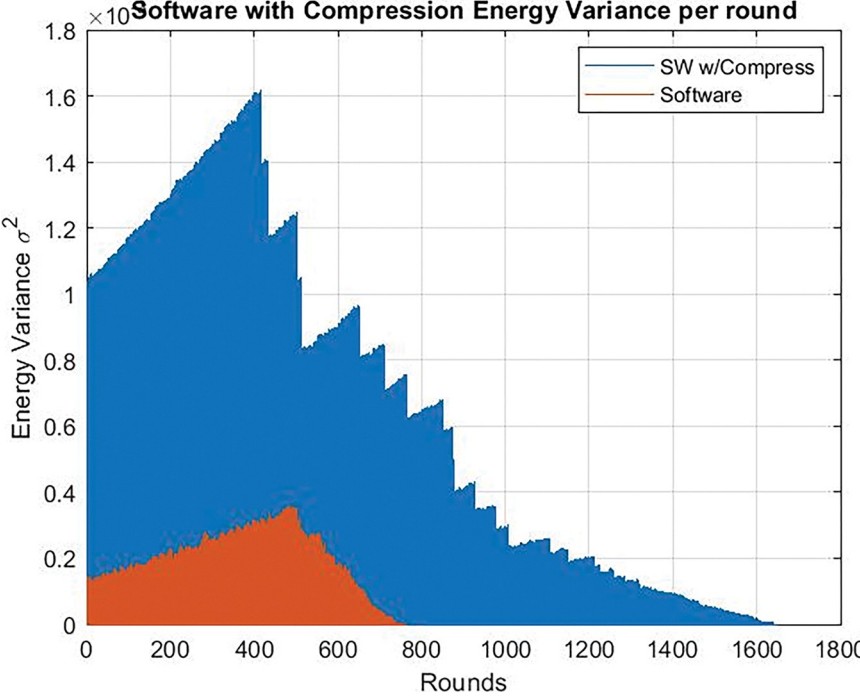

**Fig 20. Energy variance per round.** Clock frequency = 80 Mhz chaos compression implemented software (SW w/C) version / software version without chaos.

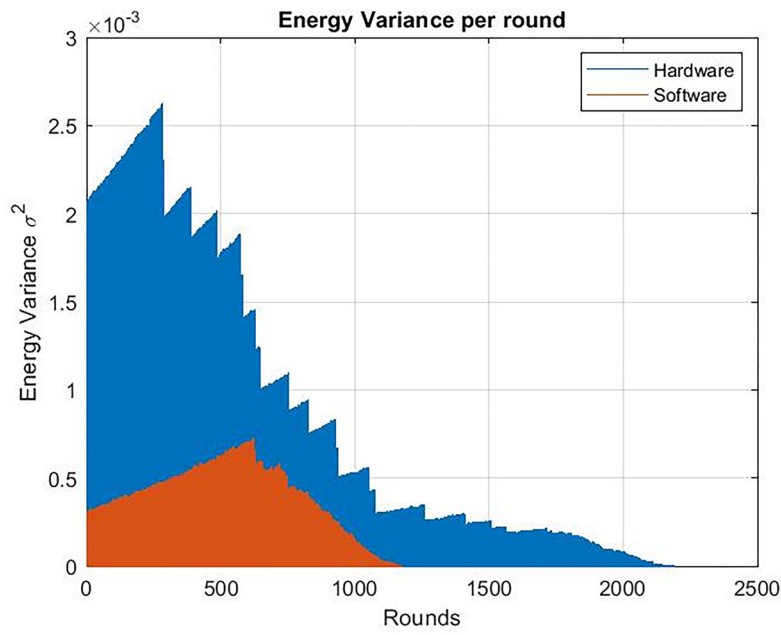

**Fig 21. Energy variance per round.** Clock frequency = 24 Mhz. Hardware version vs software without chaos.

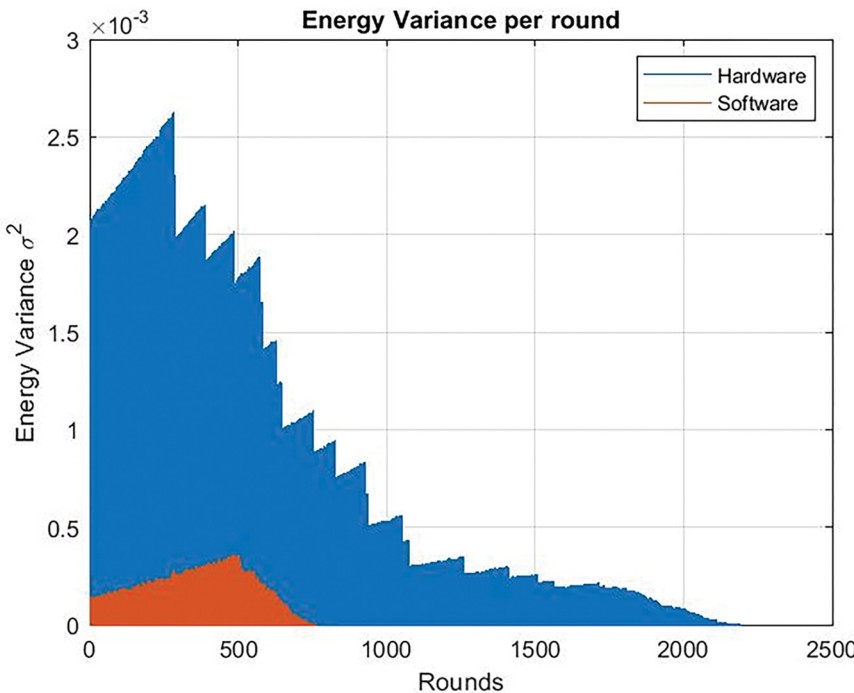

**Fig 22. Energy variance per round.** Clock frequency = 80 Mhz. Hardware version vs software without chaos.

since the high energy consumption due to the large amount of data transmitted is avoided with the compression technique applied in the dedicated circuit. With respect to [38], a system that uses a constant bit rate (CBR) mechanism to reduce energy consumption and two types of predefined nodes, some with more energy than the latter, our proposal provides more flexibility in the choice of the cluster head, since any node can adopt any role (cluster head or sensor node). Moreover, in [38] the hash is computed with the SHA256 algorithm while our proposal uses chaos hashing, a less complex hardware implementation with lower energy consumption. Finally, regarding work [40], where researchers use the LNC encryption algorithm to increase the efficiency of WSNs in terms of number of alive nodes, our proposal defines a dedicated circuit to take care of the complexities related to blockchain functions. We have also used data compression to optimize energy consumption, in addition to using the compression process as an algorithm for data encryption. With this process we have obtained twice the power.

Once it is clear that a blockchain can be implemented in a WSN, in future works network performance under the threat of different attacks will be tested and the security enhancements of blockchain in such scenarios will be evaluated. Network behavior will be monitored under attacks and it will be studied how the attack will be detected by a trust mechanism based on the dedicated circuit using chaos technology as well as the proposed lightweight encryption. Additionally, we will compare our technique with blockchain implementations in other microcontroller families, mainly those working at higher frequencies.

## Author Contributions

**Conceptualization:** José Carlos Campelo.

**Formal analysis:** José Carlos Campelo.

**Funding acquisition:** Alberto Bonastre Pina.

**Investigation:** Maytham S. Jabor, Aqeel Salman Azez.

**Methodology:** Alberto Bonastre Pina.

**Resources:** Alberto Bonastre Pina.

**Software:** Maytham S. Jabor, Aqeel Salman Azez.

**Supervision:** José Carlos Campelo, Alberto Bonastre Pina.

**Writing – original draft:** Maytham S. Jabor, Aqeel Salman Azez, José Carlos Campelo, Alberto Bonastre Pina.

**Writing – review & editing:** José Carlos Campelo, Alberto Bonastre Pina.

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
