## [Decision Letter · Decision Letter 0]

23 Jan 2023

PONE-D-22-33611New approach to improve power consumption associated with blockchain in WSNsPLOS ONE

Dear Dr. Jabor,

Thank you for submitting your manuscript to PLOS ONE. After careful consideration, we feel that it has merit but does not fully meet PLOS ONE’s publication criteria as it currently stands. Therefore, we invite you to submit a revised version of the manuscript that addresses the points raised during the review process.

ACADEMIC EDITOR: Reviewers were in general agreement that the manuscript requires a revision before to be considered for publication.

Authors should carefully take into account their suggestions and comments for the acceptance.In particular, they should improve the overall presentation of the manuscript, fix English where needed, clarify better the context and motivations of the research.

Finally, in preparing their revised version, authors should follow the guide lines reported at the following address to comply with the journal requirements:

https://journals.plos.org/plosone/s/submission-guidelines

Authors also should share their data, material, and artifacts according to https://journals.plos.org/plosone/s/materials-software-and-code-sharing

We look forward to receiving your revised manuscript.

Kind regards,

Letterio Galletta

Academic Editor

PLOS ONE

Journal Requirements:

4. Please amend your authorship list in your manuscript file to include author Maytham Jabor.

5. We note that Figures 2, 9 and 10 in your submission contain copyrighted images. All PLOS content is published under the Creative Commons Attribution License (CC BY 4.0), which means that the manuscript, images, and Supporting Information files will be freely available online, and any third party is permitted to access, download, copy, distribute, and use these materials in any way, even commercially, with proper attribution. For more information, see our copyright guidelines: http://journals.plos.org/plosone/s/licenses-and-copyright.

a. You may seek permission from the original copyright holder of Figures 2, 9 and 10 to publish the content specifically under the CC BY 4.0 license. 

Reviewers' comments:

Reviewer's Responses to Questions

**Comments to the Author**

1. Is the manuscript technically sound, and do the data support the conclusions?

Reviewer #1: Yes

Reviewer #2: Yes

2. Has the statistical analysis been performed appropriately and rigorously? 

Reviewer #1: I Don't Know

Reviewer #2: N/A

3. Have the authors made all data underlying the findings in their manuscript fully available?

Reviewer #1: No

Reviewer #2: Yes

4. Is the manuscript presented in an intelligible fashion and written in standard English?

Reviewer #1: Yes

Reviewer #2: No

5. Review Comments to the Author

Reviewer #1: The author minimised the processing load of generating the blockchain hash value, encrypting and compressing the data to reduce the overall traffic and energy to add blockchain in WSNs. In the paper, they design a dedicated circuit to implement the compression technique and generate the blockchain hash values and data encryption. The compression algorithm is based on chaotic theory. Finally, they compare the power consumed by a WSN using a blockchain implementation with and without the dedicated circuit, showing that the dedicated hardware reduces the power consumption up to 63%.

SIGNIFICANCE

The idea of the paper is a significant advance in state of the art and helps researchers open new research directions.

ORIGINALITY

The idea of adding blockchain technology to Wireless Sensor Networks is not new, but they innovate it using a dedicated circuit that reduces power consumption.

TECHNICAL QUALITY

In the paper, the author prose a new way to use blockchain technology on Wireless Sensor Networks reducing several limitations, such as energy and memory size. To overcome these problems, they design a specific circuit. They provide a comparison of the power consumption using a blockchain implementation with and without the dedicated circuit using MATLAB. The results show that the dedicated hardware reduces power consumption by up to 63%.

In my opinion, the author should better discuss their result to stress what are the advantages of using blockchain. For example, they could insert some results about the network's resilience to churn nodes.

In the Introduction, the authors said that speed is one of the advantages of the blockchain. Could they explain why? Usually, the transaction speed is one of the blockchain limitations.

In Section Proposed System, the authors talk about the POS consensus algorithm. Is it proof of stake? It needs to be clarified how they use it.

The author should better comment on Figures 15 and 16 to stress the advantages of their model compared to the others.

The Authors should share their data and codes to allow everyone to reproduce their results.

SCHOLARSHIP AND QUALITY OF WRITING

This paper is well-written and organized.

The paper situates the work concerning the state of the art, citing and comparing the other papers about WSNs and blockchain.

Some other comments:

- [30],[31],[35] are websites and are better as footnotes.

-The quality of Figures 3, 6, 7, 8, and 9 has to be increased.

- The arrow size of Figure 2 should be increased.

- Figure 5 is cited before Figure 4; they should be inverted.

- Section Circuit design and implementation should start with an introduction and not directly with a subsection.

Reviewer #2: 

#SUMMARY

The manuscript applies blockchain technology (BC) to Wireless Sensor Networks (WSNs) in order to add a certain level of security (confidentiality and integrity), avoiding a centralization approach.

A typical issue with BC is that it requires a certain amount of computation capabilities and energy that are critical in WSNs.

To face this issue, the authors propose an energy minimization strategy that minimizes the processing load of the BC hash value and encrypts and compresses data that travels from the cluster heads to the base station reducing the general traffic.

The main novelty of the manuscript is that the compression algorithm used to transfer information between nodes of the network is based on chaotic theory.

Authors argue that compressing data using chaotic theory has at least two advantages. The first is that it can be easily implemented in hardware. The second one is that it allows the creation of low-power, cheap signals and can be added in the small areas of electronic circuits.

To support their design, the authors provide a hardware implementation of their proposal in Verilog implementation and run it on an FPGA.

They perform an experimental evaluation showing a reduction of 63% in energy consumption.

#EVALUATION

The topic of the paper is interesting and within the scope of the journal. The paper aims at providing an effective energy consumption associated with blockchain in Wireless Sensor Networks with the use of a dedicated circuit.

We welcome the goal of the paper but we believe that the paper at the current stage is not ready for publication yet. For this reason, we suggest that the manuscript undergoes a major revision.

We identify two kinds of issues that should be addressed in future submissions.

The first concerns the overall presentation throughout the paper:

- a clear definition of security property should be added It should be analyzed whether the level of security is increased, which security properties are provided, and the overall benefits from a security point of view.

- citations of relevant concepts are missing (i.e. the form “Most researchers” must be followed by the associated references, “Leach protocol” in the “Proposed system” section, “Verilog” in the “Circuit design” section) or a not correctly placed (i.e. the 8th source)

- several sentences are too long and this affects the understanding of the concepts/ideas

- certain concepts that are repeated

- the use of “This way” should be dramatically reduced

- the use of brackets should be limited. They should not contain important concepts (i.e. “the hash of the current block” in the “Introduction” section)

- the English should be improved (there are several grammatical and punctuation errors that should be avoided)

- the style of the article is not coherent (so bulleted lists, approaches name, and acronyms should be written in the same way)

- the advantages of a concept (such as blockchain technology, or chaos theory) should be inserted after its definition. The exchange will increase the clarity of the paper and its basic notion.

basic concepts on BC and chaotic theory could be introduced in a suitable “Background” section.

Below we provide a list of the issue for each section that should be addressed:

- In “Abstract”:

- it should be interesting to have a clear motivation about how this approach added confidentiality and integrity. (It could be inserted in one of the sections of the paper but at the current stage is missing.)

- In “Introduction:

- some ideas are unclear (such as the adoption of BC in WSNs and why it is a successful strategy in the use of chaos on hardware)

- Nakamoto’s paper should be cited when introducing the notion of BC certain definitions are missing. In particular, we suggest introducing the blockchain as a distributed ledger because this concept is used in the BC advantages and in the “Proposed system” section.

- In “State of the art”:

- the illustrated existing approaches must achieve energy consumption (as the introductory sentence explains) but the 25th source is inserted even if it does not fulfill this goal. Its inclusion should be motivated.

- all the sources should be classified according to some criteria that are a priori defined (i.e. including implemented methodology/ies, security properties provided, innovation/s) and it could be useful to insert a table that summarizes them.

- all the existing approaches should be compared with the researchers' one as it is done for the 28th source. If the 28th source is more relevant than the others should be motivated.

- In “Proposed System”:

- the use of the figures is not effective (such as Fig.5 is cited in advance compared to its position)

- some sentences could be simplified (such as “Nevertheless, …., encrypted”)

- the layout associated with the data flowing into the network should be improved. Fig.3 should be inserted before the overall description of the mechanism to better understand it.

- In “Circuit design and implementation”:

- the introductory sentence that illustrates the purpose of the section is missing

- the overall structure should be simplified because there are a lot of repetitions (such as the definition of the parameters should be listed once). Some basic ideas are already expressed in the “Introduction” section so are redundant.

- the Algorithm definition and analysis are not clear (such as the parameters could be better visualized using a bulleted list)

- there is a line spacing issue after Algorithm 3 and the explanation of this Algorithm could be improved.

- the interpolation part should be rephrased and clarified when and why it is required to be used (the use of an example, 37th source, is not enough)

- In “Experimentation and Results”:

- the acronym CS is incorrectly used. It stands for compressed sensing, not for signal compression. The use of the same acronym is misleading.

- Fig. 9 and Fig. 10 could be replaced with tables showing the outcomes of the FPGA implementation to have clear evidence of the values.

- the analysis of how the microcontroller works at higher frequency should be added in the “Future work” section

- Figures should be deployed and analyzed. In particular, Fig. 15 and Fig.16 show three different approaches, their results should be examined and the benefits (or the drawbacks) should be illustrated.

- In “Conclusions and future work”:

- the definition of which type of attack will be analyzed as future work should be added

6. PLOS authors have the option to publish the peer review history of their article (what does this mean?). If published, this will include your full peer review and any attached files.

Reviewer #1: No

Reviewer #2: No

---

## [Author Response · Author response to Decision Letter 0]

17 Mar 2023

Dear Reviewer's

thanks for evaluating this article. Your comments have been very useful to improve our paper. The revised manuscript includes your suggestions

---

## [Decision Letter · Decision Letter 1]

5 May 2023

New approach to improve power consumption associated with blockchain in WSNs

PONE-D-22-33611R1

Dear Dr. Jabor,

We’re pleased to inform you that your manuscript has been judged scientifically suitable for publication and will be formally accepted for publication once it meets all outstanding technical requirements.

Kind regards,

Letterio Galletta

Academic Editor

PLOS ONE

Reviewers' comments:

Reviewer's Responses to Questions

**Comments to the Author**

1. If the authors have adequately addressed your comments raised in a previous round of review and you feel that this manuscript is now acceptable for publication, you may indicate that here to bypass the “Comments to the Author” section, enter your conflict of interest statement in the “Confidential to Editor” section, and submit your "Accept" recommendation.

Reviewer #1: All comments have been addressed

Reviewer #2: All comments have been addressed

2. Is the manuscript technically sound, and do the data support the conclusions?

Reviewer #1: Yes

Reviewer #2: Yes

3. Has the statistical analysis been performed appropriately and rigorously? 

Reviewer #1: Yes

Reviewer #2: Yes

4. Have the authors made all data underlying the findings in their manuscript fully available?

Reviewer #1: Yes

Reviewer #2: Yes

5. Is the manuscript presented in an intelligible fashion and written in standard English?

Reviewer #1: Yes

Reviewer #2: Yes

6. Review Comments to the Author

Reviewer #1: The authors have adequately addressed all my comments raised in a previous round of review, and I think this manuscript is now acceptable for publication.

Reviewer #2: The topic of the paper is interesting and within the scope of the journal. It aims at providing an effective energy consumption associated with blockchain in Wireless Sensor Networks with the use of a dedicated circuit.

The authors have significantly improved the manuscript by taking into account the proposed changes. All the critical issues identified in the first revision have been addressed.

The paper is complete from a stylistic and conceptual point of view at the current stage so we believe that is ready for publication.

7. PLOS authors have the option to publish the peer review history of their article (what does this mean?). If published, this will include your full peer review and any attached files.

Reviewer #1: No

Reviewer #2: No

---

## [Editor Report · Acceptance letter]

11 May 2023

PONE-D-22-33611R1 

New approach to improve power consumption associated with blockchain in WSNs 

Dear Dr. Jabor:

I'm pleased to inform you that your manuscript has been deemed suitable for publication in PLOS ONE. Congratulations! Your manuscript is now with our production department. 

Kind regards, 

on behalf of

Dr. Letterio Galletta 

Academic Editor

PLOS ONE